# The locus coeruleus broadcasts prediction errors across the cortex to promote sensorimotor plasticity

Rebecca Jordan[1][\*][†], Georg B Keller[1,2]

[1]Friedrich Miescher Institute for Biomedical Research, Basel, Switzerland; [2]Faculty of Sciences, University of Basel, Basel, Switzerland

**\*For correspondence:**
rjordan3@ed.ac.uk

**Present address:** [†]Simons Initiative for the Developing Brain, University of Edinburgh, Edinburgh, United Kingdom

**Competing interest:** The authors declare that no competing interests exist.

**Abstract** Prediction errors are differences between expected and actual sensory input and are thought to be key computational signals that drive learning related plasticity. One way that prediction errors could drive learning is by activating neuromodulatory systems to gate plasticity. The catecholaminergic locus coeruleus (LC) is a major neuromodulatory system involved in neuronal plasticity in the cortex. Using two-photon calcium imaging in mice exploring a virtual environment, we found that the activity of LC axons in the cortex correlated with the magnitude of unsigned visuomotor prediction errors. LC response profiles were similar in both motor and visual cortical areas, indicating that LC axons broadcast prediction errors throughout the dorsal cortex. While imaging calcium activity in layer 2/3 of the primary visual cortex, we found that optogenetic stimulation of LC axons facilitated learning of a stimulus-specific suppression of visual responses during locomotion. This plasticity – induced by minutes of LC stimulation – recapitulated the effect of visuomotor learning on a scale that is normally observed during visuomotor development across days. We conclude that prediction errors drive LC activity, and that LC activity facilitates sensorimotor plasticity in the cortex, consistent with a role in modulating learning rates.

## eLife assessment

This **important** study provides **convincing** evidence that locus coeruleus is activated during visuomotor mismatches. Gain of function optogenetic experiments complement this evidence and indicate that locus coeruleus could be involved in the learning process that enables visuomotor predictions. This study, therefore, sets the groundwork for the circuit dissection of predictive signals in the visual cortex. Loss-of-function experiments would strengthen the evidence of the involvement of locus coeruleus in prediction learning. These results will be of interest to systems neuroscientists.

## Introduction

Through experience with the world, brains learn to predict the sensory feedback generated by movement. For instance, learning the precise relationship between motor commands and the resulting visual feedback is the basis for both feedback control of movements as well as distinguishing self-generated from externally generated sensory feedback. One way that the brain may learn to do this is via predictive processing (*Keller and Mrsic-Flogel, 2018*; *Rao and Ballard, 1999*). In a sensorimotor version of this framework, the brain learns an internal model which transforms motor activity into a prediction of the sensory consequences. The framework postulates that specific neurons are responsible for computing discrepancies – known as prediction errors – between the prediction and the actual sensory inputs received. The neocortex has been shown to compute signals consistent with sensorimotor prediction errors (*Eliades and Wang, 2008*; *Keller and Mrsic-Flogel, 2018*; *Schneider*

*et al., 2018*). In the primary visual cortex (V1), excitatory neurons in layer 2/3 compute mismatches between visual flow speed and locomotion speed (*Attinger et al., 2017*; *Jordan and Keller, 2020*; *Keller et al., 2012*; *Zmarz and Keller, 2016*). Such sensorimotor prediction error responses in the cortex depend on ongoing and developmental experience with sensorimotor coupling (*Attinger et al., 2017*; *Schneider et al., 2018*; *Vasilevskaya et al., 2022*; *Widmer et al., 2022*), and therefore the computation underlying them is actively learned. In predictive processing, one main function of prediction errors is to drive corrective plasticity in the internal model that generates predictions (*Keller and Mrsic-Flogel, 2018*).

During the first sensorimotor experience in life, prediction errors should strongly drive learning of an internal model of the world, while later in life, prediction errors should update internal models primarily when overall prediction errors are abundant (for instance, in novel or volatile situations). One way to implement such a developmental or contextual shift in the amount of plasticity induced by a particular prediction error would be a gating signal. The characteristics of a gating signal are that it is activated by prediction errors and that it permits increases in plasticity. In addition, because it is not driving plasticity, but simply gating it, it can be low dimensional (i.e. the same for all neurons). Prime candidates for a gating signal are neuromodulatory systems. Dopaminergic neurons in the midbrain signal reward prediction errors (*Kim et al., 2020*; *Schultz, 2016*) and action prediction errors (*Greenstreet et al., 2022*), which are thought to gate plasticity in downstream targets to support learning of stimulus-reward associations and goal directed behavior (*Cox and Witten, 2019*; *Flagel et al., 2011*; *Reynolds et al., 2001*; *Steinberg et al., 2013*). Indeed, the catecholamines – dopamine and noradrenaline – regulate and gate plasticity in the cortex (*Choi et al., 2005*; *He et al., 2015*; *Seol et al., 2007*). However, the midbrain dopaminergic system projects only sparsely to most of the sensory neocortex of the mouse, including V1 (*Nomura et al., 2014*), where visuomotor predictions are thought to be learned (*Attinger et al., 2017*; *Widmer et al., 2022*). Instead, the major catecholaminergic input to the cortex is the locus coeruleus (LC). LC activity has primarily been investigated in the context of classical stimulus presentations and reinforcement tasks. Such studies have shown that the LC responds to many different events, including unexpected task outcomes (*Bouret and Sara, 2004*; *Breton-Provencher et al., 2022*), novelty (*Takeuchi et al., 2016*; *Vankov et al., 1995*), and unexpected sensory stimuli in general (*Aston-Jones and Bloom, 1981*; *Deitcher et al., 2019*; *Foote et al., 1980*; *Hervé-Minvielle and Sara, 1995*). In zebrafish, noradrenergic neurons increase their activity during the prolonged visuomotor mismatches of failed swim attempts to drive 'giving up' behavior (*Mu et al., 2019*). We hypothesized that if the mouse LC is activated during similar visuomotor mismatches, this could represent visuomotor prediction error signaling that would function to gate prediction error driven plasticity in output circuits like the cortex.

To test fundamental aspects of this idea, we imaged calcium activity in LC axons in the cortex during visuomotor mismatches, and optogenetically manipulated LC activity levels during visuomotor experience to probe for experience-dependent changes in cortical activity. We found that LC axons in both sensory and motor cortical regions respond to unsigned visuomotor prediction errors, that is, unpredicted visual motion or unpredicted visual halts during locomotion. We then combined optogenetic LC axon stimulation with two-photon calcium imaging of layer 2/3 neurons in V1, to show that the LC greatly facilitates a form of plasticity consistent with predictive visuomotor learning: learned suppression of nasotemporal visual flow responses during forward locomotion. These results support the idea that a key function of the LC is to facilitate prediction-error-driven cortical plasticity.

## Results

We first assessed whether locus coeruleus (LC) axons convey sensorimotor prediction errors to the cortex. To record LC axonal activity, we expressed axon-targeted, Cre-dependent GCaMP6s in noradrenergic neurons of the LC via stereotactic viral vector injections in 13 NET-Cre mice (NET = norepinephrine transporter; *Wagatsuma et al., 2018*). We confirmed that these injections labelled only cells in the LC, and that GCaMP6s positive cells were also immunoreactive for tyrosine hydroxylase, a marker for catecholaminergic neurons (*Figure 1—figure supplement 1A*). For two-photon imaging, we implanted a 4 mm imaging window either over the right V1 and surrounding cortical regions (posterior cortex, 9 mice), or over the right primary and secondary motor cortices (anterior cortex, 4 mice). Prior to imaging, mice were habituated to a virtual reality (VR) system consisting of a virtual tunnel, with walls patterned with vertical sinusoidal gratings (*Figure 1A*). In the closed loop condition,

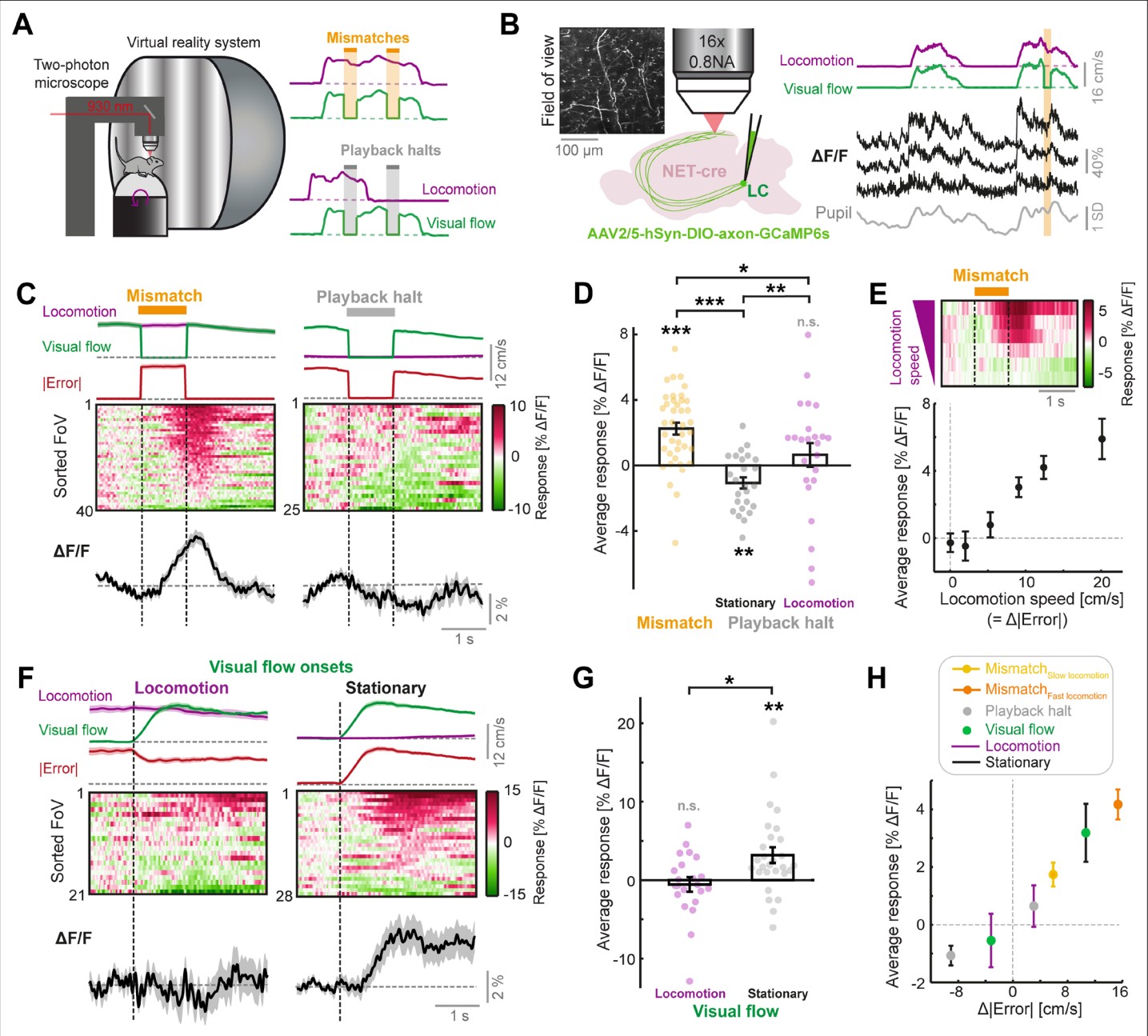

**Figure 1.** LC axonal calcium activity reflects unsigned visuomotor prediction errors. (**A**) Left: Schematic of the two-photon microscope and virtual reality system. Right: Schematics of the two virtual reality conditions. Top: closed loop condition in which visual flow speed (green) is yoked to locomotion speed (purple). Mismatches consist of 1 s halts in visual flow during locomotion (orange shading). Bottom: open loop condition in which visual flow from the closed loop session is replayed to the mouse uncoupled from locomotion. Playback halts are 1 s halts in visual flow in the open loop condition (gray shading). (**B**) Two-photon calcium imaging of LC axons in the dorsal cortex. Left: NET-Cre mice underwent stereotactic viral vector injections to express axon-targeted GCaMP6s in the LC. Four weeks later, axons were imaged in two regions of the cortex. Inset shows example two-photon field of view (FoV). Right: Example ΔF/F traces (black) from three different axon segments in the same FoV, recorded during the closed loop condition. Purple and green traces show locomotion and visual flow speed respectively. Orange shading indicates mismatch events. Gray trace shows pupil diameter. (**C**) Responses of LC axons to visual flow halt stimuli (left: mismatches, and right: playback halts while the mouse is stationary). Top: Locomotion speed (purple), visual flow speed (green) and absolute error (red; absolute difference between visual flow speed and locomotion speed) each averaged across trials. Middle: Heat map of the average responses of different fields of view (FoVs) sorted according to response magnitude. Bottom: Response averaged across FoVs. Shading indicates SEM. (**D**) Average responses per FoV to mismatches and playback halts, while the mouse was either stationary or locomoting. Error bars indicate SEM across FoVs. Here and in other panels, n.s.: not significant, *: p<0.05, **: p<0.01, ***: p<0.001. For complete statistical information see ***Supplementary file 1***. (**E**) Mean mismatch response of LC axons averaged across FoVs as a function of locomotion speed.

*Figure 1 continued on next page*

*Figure 1 continued*

Top: Heat map of the average mismatch response, sorted according to locomotion speed (speeds correspond to below plot). Bottom: Mean and SEM (error bars) of the mismatch responses averaged over FoVs as a function of locomotion speed. Only datapoints with at least 10 FoVs in each speed bin are included. (**F**) Responses of LC axons to visual flow onsets in the open loop condition during locomotion (left) or stationary periods (right), plotted as in panel **C**. (**G**) Average responses of LC axons to visual flow onsets while the mouse was either locomoting or stationary, plotted as in panel **D**. (**H**) Responses of LC axons to the different visual flow stimuli in different conditions averaged across FoVs, color coded according to the legend, and plotted as a function of the change in absolute error between locomotion speed and visual flow speed during the stimulus (relative to 1 s prior to stimulus onset). Error bars indicate SEM.

The online version of this article includes the following figure supplement(s) for figure 1:

**Figure supplement 1.** Additional information and analyses on two-photon imaging of LC axons.

---

locomotion was coupled to visual movement of the tunnel, and in open loop conditions, the walls moved independent of mouse locomotion. After habituation to head-fixation, we began imaging across up to five non-overlapping fields of view (FoV) in each imaging window, yielding a total of 31 posterior cortex FoVs and 9 anterior cortex FoVs. Axonal fluorescence showed strong correlations with pupil diameter (*Figure 1—figure supplement 1B*) typical for LC axons (*Reimer et al., 2016*), rapid responses to air puffs (*Figure 1—figure supplement 1C*) as previously reported (*Breton-Provencher et al., 2022*; *Deitcher et al., 2019*), and increases in fluorescence at locomotion onset that decayed during the locomotion bout (*Figure 2—figure supplement 1*) consistent with earlier work (*Reimer et al., 2016*). We found that within a FoV, individual axon segments exhibited highly correlated activity (*Figure 1B* and *Figure 1—figure supplement 1D–F*). Given that the algorithms that identify axons (i.e. determine whether two axon segments belong to the same axon) rely on temporal correlations of activity (*Leinweber et al., 2017*; *Mukamel et al., 2009*), axon segmentation is difficult in these data. Thus, to prevent oversampling of the data, we pooled data from all axon segments within a FoV for further analysis.

## Locus coeruleus cortical axons signal unsigned visuomotor errors

To determine whether LC axons respond to visuomotor errors, we first measured their responses to visuomotor mismatch stimuli (unexpected 1 s halts of visual flow during locomotion in the closed loop condition). Indeed, LC axons showed significant increases in fluorescence following mismatch presentations (*Figure 1C–D*). Assuming these responses reflect visuomotor prediction errors, we would expect to find two things:

First, the responses should depend on precise coupling between visual flow speed and locomotion speed (i.e. when visual flow is predictable from locomotion). Responses to visuomotor mismatch in closed loop conditions should be higher than responses to a replay of the same visual flow halt (termed 'playback halt') in open loop conditions, even when the mouse is locomoting. Playback halts are visually identical to mismatches but occur independent of whether the mouse is stationary or locomoting (*Figure 1A*). Responses of LC axons to playback halts when the mouse was stationary were negative and substantially different from mismatch responses (*Figure 1C–D*), demonstrating that the response of LC axons to visuomotor mismatch cannot be explained by the visual stimulus alone. The mismatch responses also could not be explained by a locomotion driven increase in gain of the visual halt response: (a) The average playback halt response when the mouse was stationary was negative, (b) there was no correlation between playback halt responses and mismatch responses across different FoVs ($R^2$=0.04, p=0.31, 25 FoVs, linear regression), and (c) mismatch responses were significantly larger than playback halt responses that occurred during locomotion in the open loop condition (*Figure 1D*). This indicated that responses in LC axons were acutely dependent on visuomotor coupling, since the only difference between a mismatch stimulus and a playback halt during locomotion is that the former occurs during visuomotor coupling and the latter does not.

Second, the mismatch response should scale with the size of the error between visual flow speed and locomotion speed, since the latter is a proxy for the prediction of visual flow speed. Since visual flow speed is zero during mismatch, and visual flow is perfectly coupled to locomotion prior to mismatch, the visuomotor error is proportional to the locomotion speed during mismatch (*Figure 1C*). Indeed, we found that LC axonal mismatch responses increased monotonically with the locomotion speed of the mouse (*Figure 1E*). Thus, LC axonal responses to visuomotor mismatch are consistent with a visuomotor prediction error signal.

Visuomotor mismatches evoke negative prediction errors because they are a condition where there is *less* visual flow than expected. However, these are not the only type of prediction error in our paradigm. In the open loop condition, visual flow onsets result in positive prediction errors (i.e., a condition with *more* visual flow than expected), especially when the mouse is stationary. We thus assessed the responses of LC axons to visual flow onsets in the open loop condition (i.e. uncoupled from locomotion). We found that LC axons showed significant responses to visual flow onsets, but only when the mouse was stationary (*Figure 1F–G*). Visual flow onsets during locomotion did not evoke significant responses on average. Altogether, LC axonal responsiveness to visual and visuomotor stimuli was best explained by the absolute (i.e. unsigned) error between locomotion and visual flow speeds ($|Error|=|Speed_{locomotion} - Speed_{visual\ flow}|$). Changes in the absolute error resulted in corresponding changes in the activity in LC axons (*Figure 1C and F*), and responses scaled with the size of $\Delta|Error|$ (i.e. the change in unsigned error magnitude, see Methods) across the different stimulus conditions (*Figure 1H*).

## Noradrenaline broadcasts visuomotor prediction errors across the dorsal cortex

Individual LC neurons can preferentially target different cortical areas (*Chandler et al., 2019*; *Kebschull et al., 2016*), but it is unclear whether these projection-specific neurons have differing response profiles. To assess the heterogeneity of LC signaling across the cortex, we first compared the calcium responses of LC axons between two imaging window locations: the anterior cortex (the primary and secondary motor cortex), and the posterior cortex (V1 and surrounding regions; *Figure 2A*). We found very similar responses to visuomotor mismatches, playback halts (stationary) and visual flow onsets (stationary; *Figure 2B–C*), as well as air puffs and locomotion onsets at these two locations (*Figure 2B and D*), and there were no significant differences in the responses between the two locations. These results are consistent with the idea that the LC sends similar visuomotor and locomotion related signals to distinct regions of the cortex, in line with recent findings showing that LC axons in different cortical regions show responses to the same sensory stimuli (*Deitcher et al., 2019*).

## Locus coeruleus axon stimulation has only a small effect on stimulus responses in layer 2/3 of V1

What could be the function of prediction errors signaled by the LC to the cortex? One possibility is that cortical prediction errors, like mismatch responses in V1 (*Keller et al., 2012*), are enhanced by LC output in order to augment error driven learning. Indeed, a prevalent idea is that noradrenaline is a mediator of gain modulation in the cortex (*Ferguson and Cardin, 2020*), including the gain increase in V1 during locomotion (*Polack et al., 2013*). To test whether the LC may enhance prediction error responses in cortical layer 2/3, we assessed the impact of transient LC axon stimulation on cortical responses using an optogenetic strategy (*Figure 3A*). First, we injected an AAV vector encoding Cre-dependent excitatory red-light activated opsin, ChrimsonR, into the right LC of 8 NET-Cre mice, while injecting a vector encoding jGCaMP8m into the right V1 to drive non-specific neuronal expression. An imaging window was then implanted over the right V1. Mice underwent habituation to the virtual reality system before calcium imaging of V1 neurons during (1) a closed loop condition with mismatches and (2) an open loop condition with 1 s fixed-speed visual flow stimuli. During imaging, we stimulated ChrimsonR by directing a 637 nm laser through the objective into the imaging window (27 mW/mm$^2$, in 15ms pulses presented at 20 Hz). This stimulation was presented (a) alone in closed loop conditions, (b) coupled to a random 50% of mismatches (*Figure 3B*), and (c) coupled to a random 50% of open loop visual flow stimuli. To assess the visual effects of optical stimulation of ChrimsonR (*Danskin et al., 2015*) in absence of the effects of optogenetically stimulating LC axons, we also imaged from a second group of six mice treated in an identical manner, but without ChrimsonR expression in the LC (control mice). Two mice were excluded from the LC-injected group due to low density of axonal labelling with tdTomato in V1, leaving 6 ChrimsonR-expressing mice in the dataset (see Methods).

Optogenetic stimulation, when presented in isolation, resulted in dilations of the right pupil in mice expressing ChrimsonR in a manner that depended on the degree of axon labelling (*Figure 3C* and *Figure 3—figure supplement 1A*), but not in control mice. This likely resulted from antidromic activation of the LC, direct stimulation of which is known to produce strong ipsilateral pupil dilation (*Liu et al., 2017*), and increased our confidence that we were indeed driving LC axons with the parameters selected. Onset of the optogenetic stimulation laser caused a small calcium response across the

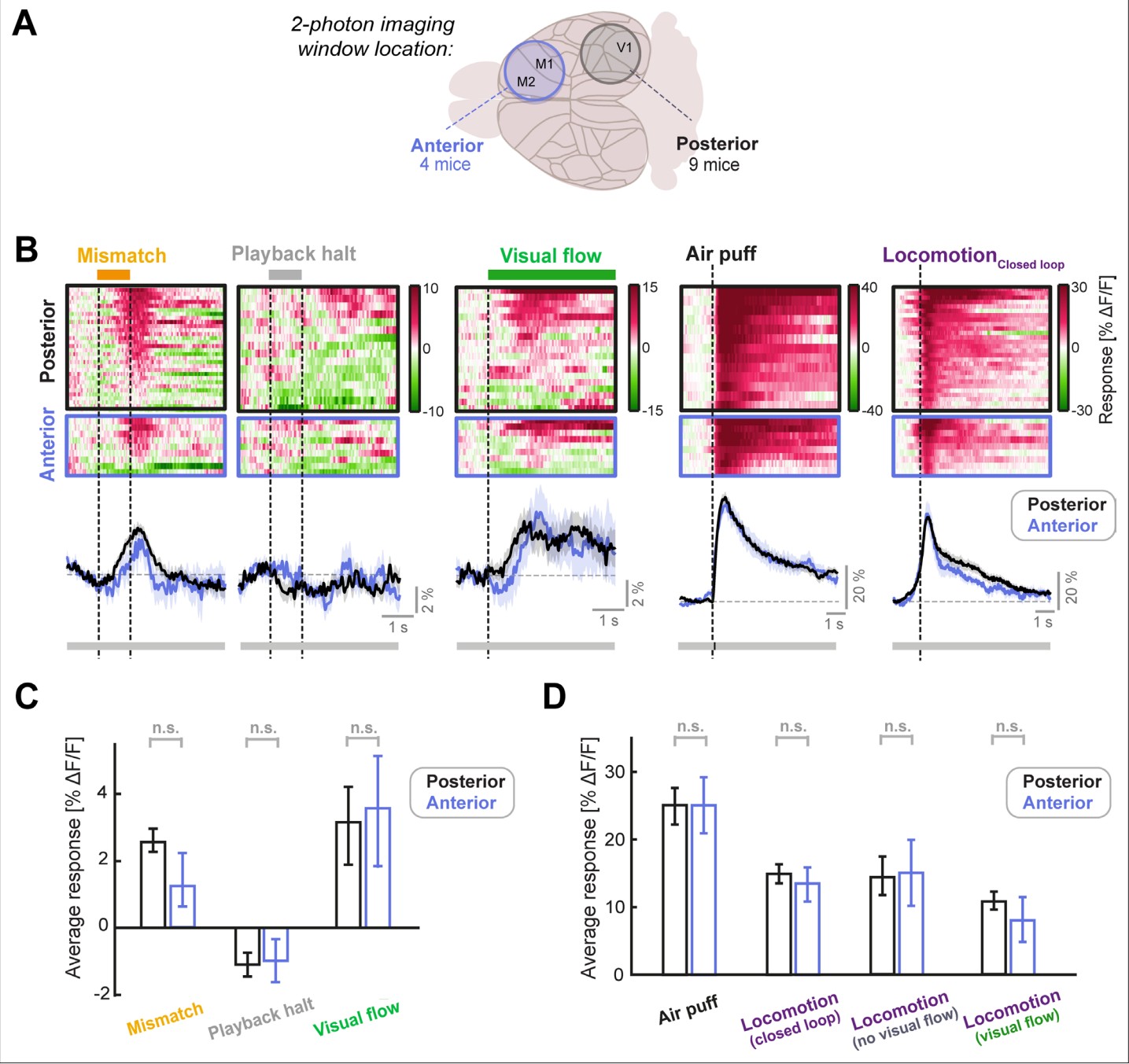

**Figure 2.** LC axonal responses are similar in sensory (posterior) and motor (anterior) cortical areas. (**A**) Schematic of the two different locations of the two-photon imaging windows used. Posterior windows were centered on V1 and included surrounding structures, while anterior windows included the primary and secondary motor cortex. (**B**) Responses of posterior and anterior FoVs to (from left to right): Mismatch, playback halts (stationary), visual flow (stationary), air puff, and locomotion onset in closed loop. Each heatmap is sorted by average response. Color scale is ΔF/F. Traces below heatmaps show average across FoVs, for posterior (black) and anterior (blue) imaging sites. Shading shows SEM. Bar below plot indicates significance of comparison at each time point using an unpaired t-test: grey indicates p>0.01, black indicates p<0.01. (**C**) Average responses of LC axons to mismatch, playback halt, and visual flow onset averaged across FoVs, compared between posterior and anterior imaging locations. Error bars indicate SEM. Here and in other panels, n.s.: not significant. For complete statistical information see **Supplementary file 1**. (**D**) As in **C**, but for air puff responses (which were infrequently used to evoke locomotion) and locomotion onset responses in various conditions, each averaged in the 3 s window after onset. Note that locomotion_Visual flow and locomotion_No visual flow occur in open loop conditions (see also **Figure 2—figure supplement 1**).

The online version of this article includes the following figure supplement(s) for figure 2:

**Figure supplement 1.** Locomotion onset responses of LC axons.

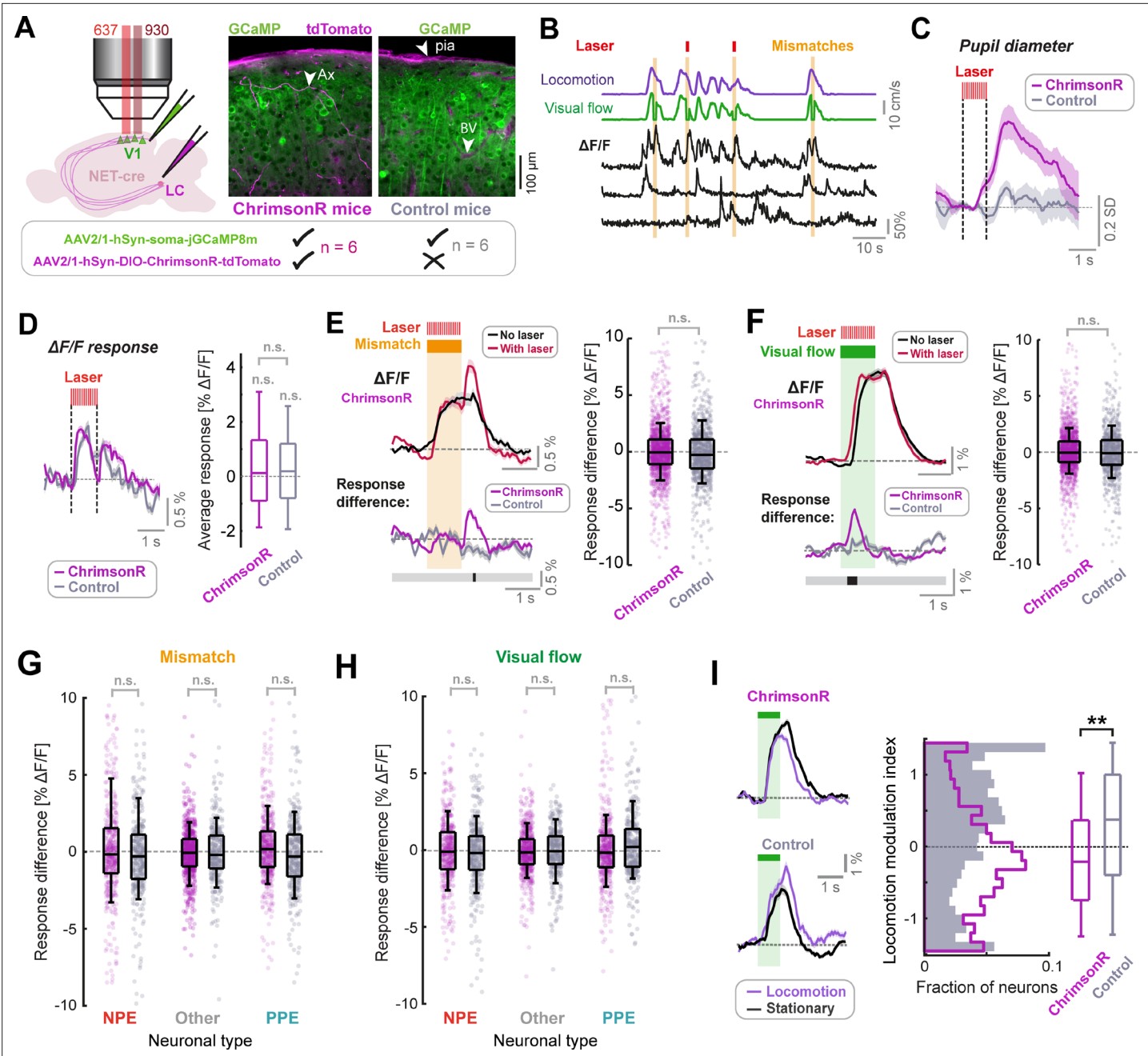

**Figure 3.** LC axon stimulation has only a small effect on stimulus responses in layer 2/3 of V1. (**A**) AAV vector injections were used to express ChrimsonR-tdTomato in LC NET-positive neurons, and jGCaMP8m in layer 2/3 neurons of V1. Simultaneous two-photon calcium imaging and optogenetic stimulation with a 637 nm laser would take place in layer 2/3. Injections into the LC were omitted in control mice. Example histology images indicate expression of jGCaMP8m (green) and tdTomato (magenta); note that blood vessels and pia mater (see arrows labelled 'BV' and 'pia' respectively) are also visible in the magenta channel. Arrow labelled 'Ax' indicates an axon. (**B**) Example ΔF/F traces for three somatic ROIs in layer 2/3 of V1 (black) and the corresponding visual flow speed (green) and locomotion speed (purple) traces. Orange shading indicates visuomotor mismatch events. Red marks indicate concurrent laser stimulation to activate ChrimsonR on a random subset of mismatch trials. (**C**) Average pupil diameter response to stimulation with the optogenetic laser presented in isolation for ChrimsonR-expressing mice (pink), and control mice (gray). Shading represents SEM. (**D**) Left: Average population ΔF/F response of layer 2/3 neurons in V1 to the onset of the optogenetic stimulation laser presented in isolation. Shading represents SEM. Right: Boxplots to compare average response (quantified in the window 0.3–1.6 s after optogenetic stimulation onset) for the two groups of mice. Here and in other panels, n.s.: not significant. For complete statistical information see **Supplementary file 1**. (**E**) Analysis of mismatch responses of all layer 2/3 neurons. Left: Average population ΔF/F response of layer 2/3 neurons in V1 of ChrimsonR-expressing mice to visuomotor mismatch either with (red) or without (black) concurrent optogenetic laser stimulation. Shading represents SEM. Below plot shows average difference between trials with and without laser stimulation averaged across the population for ChrimsonR-expressing (pink) and control (gray) mice. Bold line

*Figure 3 continued on next page*

*Figure 3 continued*

below indicates timepoints where control and ChrimsonR-expressing datasets significantly differed (black) or not (gray). Right: Boxplots to compare the difference in average mismatch response between trials with and without optogenetic laser stimulation. Pink points indicate data for each neuron from ChrimsonR-expressing mice, and gray points indicate data from control mice. Note, statistical tests against zero for the effect in ChrimsonR-expressing mice were also insignificant (see *Supplementary file 1*). (**F**) As for **E**, but for responses to 1 s fixed speed visual flow stimuli in open loop conditions. (**G**) As for boxplots in panel **E**, but for mismatch responses of three different functionally defined neuronal groups: NPE = negative prediction error neurons, with large responses to mismatches, PPE = positive prediction error neurons, with large responses to visual flow, and an intermediate group 'other'. See Methods and *Figure 3—figure supplement 2* for information on neuronal types. Note, statistical tests against zero for the effect in ChrimsonR-expressing mice were also insignificant (see *Supplementary file 1*). (**H**) As for **G**, but for responses to 1 s fixed speed visual flow stimuli in open loop conditions. (**I**) Left: Population average responses to visual flow stimuli during stationary periods (black) and during locomotion (purple), for control mice (bottom), and ChrimsonR-expressing mice (top). Shading represents SEM. Note that only trials *without* concurrent optogenetic stimulation are included here. Right: Histograms and boxplots to show distribution of locomotion modulation index for the visual responses recorded in control (gray) and ChrimsonR-expressing mice (pink). Here, **\*\***: p<0.01 (see *Supplementary file 1*).

The online version of this article includes the following figure supplement(s) for figure 3:

**Figure supplement 1.** Dependence of laser stimulation effects on ChrimsonR-labelled LC axon density.

**Figure supplement 2.** Classification of functional neuronal types in V1 layer 2/3.

population in ChrimsonR-expressing mice, which was indistinguishable from that caused in control mice (*Figure 3D*). This was likely due to a visual effect of the stimulation laser. The lack of a difference in this stimulation response between ChrimsonR-expressing mice and control mice indicated that optogenetic stimulation of LC axons did not have a direct measurable effect on calcium activity of L2/3 neurons in V1.

We then analyzed the effect of optogenetic activation of LC axons on mismatch and visual flow responses of the layer 2/3 population. To take account of visual effects of the laser, we compared the effect of the laser in ChrimsonR-expressing mice with that in control mice. Responses to visual flow onsets (i.e. at the beginning of the visual flow stimulus, or at the offset of mismatch) appeared transiently more pronounced during laser stimulation in ChrimsonR-expressing mice only (*Figure 3E–F*). This was most notable when plotting the difference in average response between trials with concurrent laser stimulation and those without: there were transient increases in response during laser-on trials at visual flow onset, and this effect was significantly different between ChrimsonR-expressing and control groups only during brief time-windows at the onset/resumption of visual flow (*Figure 3E–F*). However, this effect was small and not evident in average responses. Averaging across the full response duration, we found no significant effects of laser stimulation, both when comparing trials with and without laser stimulation in the ChrimsonR-expressing group, and when comparing the difference in response size between ChrimsonR-expressing and control mice (*Figure 3E–F*). No relationship could be found between the degree of LC axonal labelling and the effect of stimulation on average response size, so ChrimsonR expression-level was likely not the limiting factor (*Figure 3—figure supplement 1B*).

In V1, there is electrophysiological and molecular evidence for at least three different layer 2/3 pyramidal neuron types, including negative prediction error (NPE) neurons which respond to visuomotor mismatches, positive prediction error neurons (PPE) which respond to unexpected visual flow (*Jordan and Keller, 2020*; *O'Toole et al., 2022*), and an intermediate group. It is therefore conceivable that LC output could have differential effects on these groups that are masked when analyzed at the population level. To quantify potential cell-type-specific effects of LC axon stimulation, we thus split the population of neurons into three groups based on locomotion onset responses (see *Figure 3—figure supplement 2* and Methods): the group with the strongest visuomotor mismatch responses (NPE), the group with the strongest visual flow responses (PPE), and an intermediate population (Other). We found no evidence for an effect of optogenetic stimulation of LC axons on average mismatch and visual flow responses across any of these cell groups (*Figure 3G–H*). We thus concluded that transient LC axon stimulation in the awake state has only a minor direct effect on the responses of neurons in layer 2/3 of V1. These results are largely in line with recent work demonstrating a lack of effect of LC stimulation on baseline firing in cortical pyramidal neurons, and an enhancement of transient responses to afferent input (*Vazey et al., 2018*).

## Phasic LC output enhances sensorimotor plasticity in layer 2/3 of V1

If transient LC activity in the awake state does have a large impact on average layer 2/3 responses directly, what is the function of visuomotor prediction errors signaled by the LC in the cortex? The LC is thought to be involved in cortical plasticity (*Kasamatsu et al., 1985*; *Kasamatsu and Pettigrew, 1976*; *Martins and Froemke, 2015*; *Shepard et al., 2015*). Catecholamine receptors in V1 are known to modulate synaptic plasticity by activating intracellular signaling cascades and interacting with eligibility traces (*Choi et al., 2005*; *He et al., 2015*; *Nomura et al., 2014*; *Seol et al., 2007*). One plausible function of LC sensorimotor prediction errors is to gate the plasticity that underlies learning of sensorimotor predictions, for example, the prediction of visual feedback during locomotion. Primary sensory cortices can indeed learn to suppress the sensory consequences of locomotion, driving selective reductions of responses to locomotion-coupled inputs (*Schneider et al., 2018*; *Widmer et al., 2022*).

While our optogenetic stimulation of LC axons had only small measurable effects on mismatch or visual flow response size (*Figure 3*), we did notice one striking difference between ChrimsonR-expressing and control mice. In V1, there is a known increase in the size of visual responses during locomotion compared to stationary periods (*Bennett et al., 2013*; *Erisken et al., 2014*; *Niell and Stryker, 2010*; *Pakan et al., 2016*; *Polack et al., 2013*; *Zmarz and Keller, 2016*). Analyzing only trials where the optogenetic stimulation laser was not on, we found that this canonical gain increase was present for control mice but reversed in sign in ChrimsonR-expressing mice: Visual responses during locomotion were suppressed on average relative to stationary periods (*Figure 3I*). This resulted in a significantly lower locomotion modulation index (see Methods) for ChrimsonR-expressing relative to control mice (*Figure 3I*). Since episodic LC axon stimulation occurred throughout the whole imaging session (i.e. across 10 min in the closed loop condition, prior to the presentation of open loop visual flow stimuli), it is possible that the reversal of locomotion modulation index reflects a result of plasticity induced by earlier LC axon stimulation.

We wanted to directly test the idea that transient LC activation enhances plasticity that leads to a reduction in locomotion modulation index over the course of visuomotor coupling experience. To do this, we designed an experiment to assess the impact of LC axon stimulation on neuronal learning about visuomotor coupling (*Figure 4A*). We expressed jGCaMP8m in neurons in V1 of 19 mice. Twelve mice also had AAV vector injections into the LC to express ChrimsonR in NET-positive neurons, while the remaining seven underwent no injection into the LC to serve as controls for the non-optogenetic effects of the protocol. Mice were habituated to the VR as before and we then proceeded to two-photon imaging in layer 2/3 of V1. During the imaging session, mice would first be exposed to 5 min of 1 s fixed-speed open loop visual flow presentations, to quantify the gain-change of visual flow responses during locomotion (*Figure 4A*). Mice were then exposed for 5–10 min to either (a) a closed loop condition where visual flow and locomotion are coupled, or (b) an open loop replay of visual flow generated from a previous closed loop session. During this session, LC axons would be periodically optogenetically stimulated as before (15ms pulses, presented at 20 Hz for 1 s every 7 s on average) to simulate phasic LC responses (*Figure 4A*). Afterwards, the open loop visual flow presentations would be repeated to quantify how the previous LC axon stimulation affected visual responses. To assess the stimulus specificity with which visual responses changed, we presented both nasotemporal and temporonasal visual flow (mice see the former during forward locomotion in the closed loop condition; *Figure 4A*). If LC axon stimulation results in enhanced plasticity of locomotion based predictions of visual flow, we would expect the following observations: (1) Suppression of visual flow responses by locomotion is enhanced after stimulation of LC axons in closed loop conditions, (2) this suppression is specific to the type of visual flow seen during forward locomotion (nasotemporal), (3) the effect depends on ChrimsonR expression in the LC, and (4) the effect is absent when LC axons are instead stimulated during open loop conditions (i.e. the effect depends on visuomotor coupling).

Optogenetic stimulation caused a degree of pupil dilation that was strongly predictable from axonal ChrimsonR expression levels in V1 quantified postmortem ($R^2$=0.89, $p<10^{-5}$, 12 mice, linear regression) (*Figure 4B*). We split the LC-injected dataset into high ChrimsonR expression (six mice) and low ChrimsonR expression (six mice) according to V1 axonal expression level (*Figure 4B* and *Figure 4—figure supplement 1A*). Low ChrimsonR-expressing mice had optogenetically evoked pupil dilations that were not statistically distinguishable from controls, while high ChrimsonR-expressing mice had overt pupil dilations (*Figure 4B* and *Figure 4—figure supplement 1B*). For the following

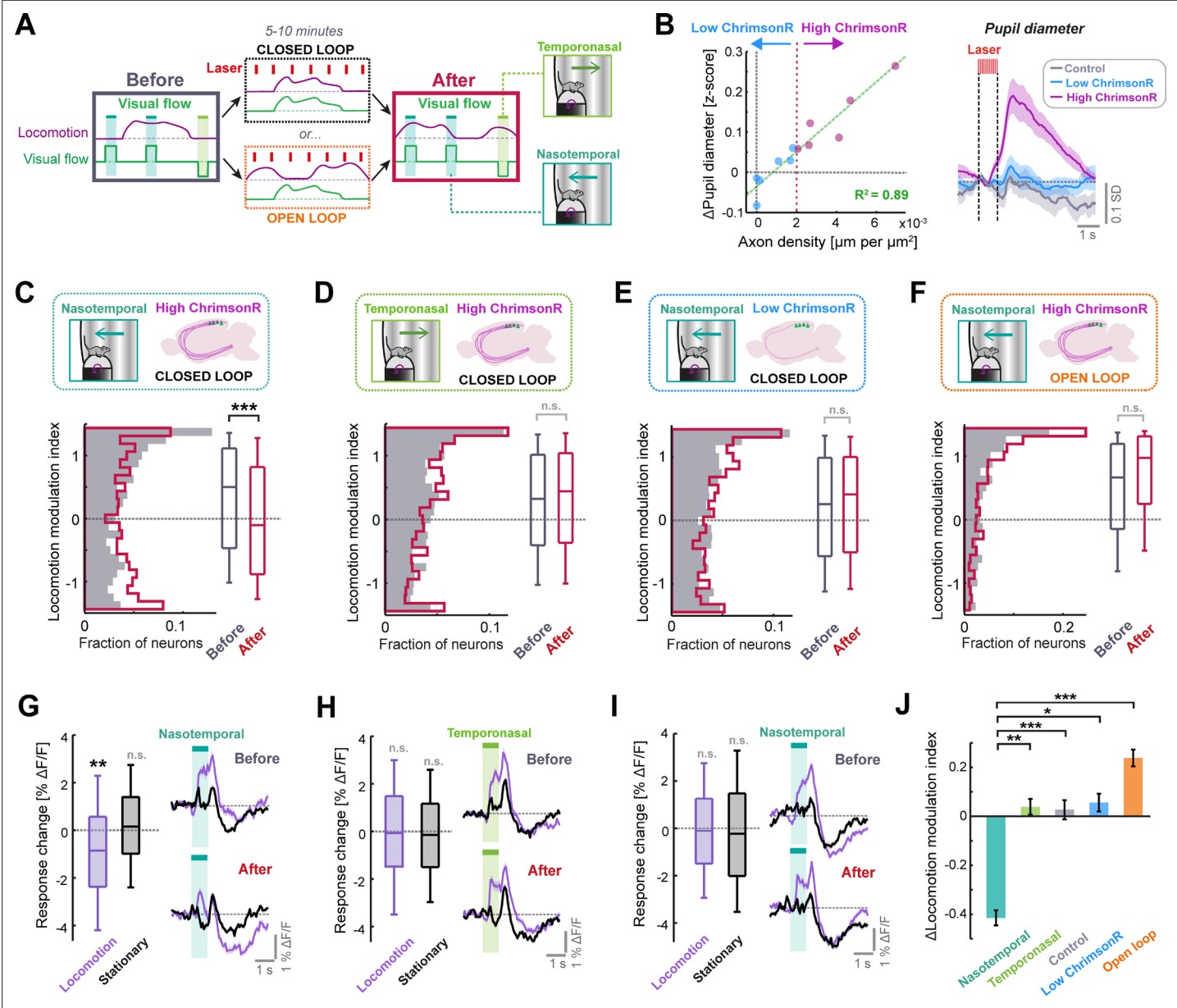

**Figure 4.** Phasic LC output enhances sensorimotor plasticity in layer 2/3 of V1. (**A**) Diagram of the experiment used to determine whether LC axon stimulation during different visuomotor coupling conditions can modulate plasticity. Visual responses are compared before and after 5–10 min of either closed or open loop conditions, during which LC axons were stimulated every 7 s on average. (**B**) Left: Scatter plot to show the relationship between density of ChrimsonR-tdTomato labelled axons in V1 (total axon length per unit area of the cortex, analyzed postmortem) and the average evoked pupil dilation during optogenetic stimulation. Green dashed line is a linear regression fit to the data, and red dotted line indicates axon density threshold used to categorize low (blue) and high (pink) ChrimsonR-expressing mice. Right: Average pupil diameter response to stimulation with the optogenetic laser for 6 mice with high ChrimsonR expression in LC axons (pink), 6 mice with low ChrimsonR expression (blue), and 7 control mice that did not receive a vector injection into the LC (black). Shading represents SEM over sessions. (**C**) Analysis of plasticity in nasotemporal visual flow responses in ChrimsonR-expressing mice undergoing optogenetic laser stimulation during closed loop visuomotor experience. Histograms and boxplots show distribution of locomotion modulation indices for the visual responses of layer 2/3 V1 neurons recorded before (dark gray) and after the stimulation period (red). Here and in all other panels: n.s.: not significant, *: p<0.05, **: p<0.01, ***: p<0.001. See **Supplementary file 1** for all statistical information. (**D**) As for panel **C**, but for temporonasal visual flow responses. (**E**) As for panel **C**, but for low ChrimsonR-expressing mice. (**F**) As for panel **C**, but for the condition in which optogenetic stimulation occurred during open loop replays of visual flow (i.e. no visuomotor coupling). (**G**) Analysis of plasticity in nasotemporal visual flow responses in high ChrimsonR-expressing mice undergoing optogenetic laser stimulation during the closed loop condition. Left: Boxplots of the change in response to visual flow after optogenetic stimulation for responses recorded during locomotion (purple) and those recorded during stationary periods (black). Right: Average population response to visual flow during locomotion (purple) and during stationary periods (black), before (top) and after (bottom) optogenetic stimulation. Shading indicates SEM. (**H**) As for panel **G**, but for temporonasal visual flow responses.

*Figure 4 continued on next page*

*Figure 4 continued*

(I) As for panel **G**, but for low ChrimsonR-expressing mice. (J) Change in locomotion modulation index after optogenetic stimulation, for the four sets of experiments from panels **C** (nasotemporal), **D** (temporonasal), *Figure 4—figure supplement 1C* (Control), **E** (low ChrimsonR expression), and **F** (open loop) (presented in the respective sequence).

The online version of this article includes the following figure supplement(s) for figure 4:

**Figure supplement 1.** Additional information and analyses on optogenetic stimulation of LC axons to drive plasticity.

**Figure supplement 2.** Differences in locomotion during stimulation cannot explain differences in the change of locomotion modulation index between groups.

analyses, we excluded 8 (5) of 49 (25) sessions from the ChrimsonR (control) dataset due to lack of locomotion during those sessions (see Methods). At the beginning of the imaging session, prior to optogenetic stimulation, neurons in layer 2/3 showed a pronounced positive locomotion modulation index for both the nasotemporal and temporonasal visual flow responses, both in high and low ChrimsonR-expressing mice and in control mice (*Figure 4C–E* and *Figure 4—figure supplement 1D*). Quantifying visual flow responses in high ChrimsonR-expressing mice after optogenetic stimulation during closed loop visuomotor experience revealed a significant reduction in the locomotion modulation index for nasotemporal visual flow responses (*Figure 4C*). This originated from a reduction in the response to visual flow during locomotion, while responses during stationary periods remained stable (*Figure 4G*). This effect was specific to the direction of visual flow seen during forward locomotion, since locomotion modulation index was preserved for temporonasal visual flow responses (*Figure 4D*), with negligible changes in response size both during locomotion and stationary periods (*Figure 4H*). The effect was also specific to high ChrimsonR-expressing mice, as locomotion modulation index did not significantly change in both control (*Figure 4—figure supplement 1D*) and low ChrimsonR-expressing mice (*Figure 4E and I*).

The selective reduction in locomotion modulation index of nasotemporal visual flow responses in high ChrimsonR-expressing mice could be due to enhanced learning to suppress reafferent visual feedback seen during forward locomotion, or it could simply be due to general exposure to nasotemporal visual flow during optogenetic stimulation (e.g., enhanced adaptation). In the latter case, response size reduction should be the same if optogenetic stimulation takes place during open loop replay of the same visual flow seen during closed loop conditions. To assess this, we repeated the same experiment, but instead of stimulating LC axons during the closed loop condition, we stimulated LC axons during open loop replays of visual flow from previous closed loop sessions of the same mice (*Figure 4A*). Instead of a stimulus selective reduction in locomotion gain, this paradigm drove a nonsignificant increase in locomotion modulation index for both nasotemporal (*Figure 4F* and *Figure 4—figure supplement 1F*) and temporonasal visual flow (*Figure 4—figure supplement 1E*) responses. This confirmed that the reduction in locomotion modulation index after stimulation of LC axons in the closed loop condition (*Figure 4C*) depends on visuomotor coupling experience and is not simply due to exposure to nasotemporal visual flow.

We conclude that LC axon stimulation over the course of minutes increases locomotion driven suppression of visual flow responses in layer 2/3 in a manner that is specific to reafferent stimulus properties and dependent on both ChrimsonR expression and visuomotor coupling (*Figure 4J*). Importantly, the effect could not be explained by differences in locomotion behavior between any of the groups analyzed (*Figure 4—figure supplement 2*).

## LC axon stimulation over minutes recapitulates visuomotor plasticity seen over days

In the virtual reality environment, mouse movement and visual flow are constrained to the forward-backward dimension, and the visuomotor gain is constant: changes in visuomotor gain that would usually occur during changes in environment geometry are absent. Mice head-fixed in the VR therefore experience a vastly reduced diversity of visuomotor coupling compared to freely moving conditions. The reduction in locomotion modulation index resulting from LC axon stimulation (*Figure 4C*) may therefore result from overlearning of this simplified visuomotor coupling.

If this interpretation is correct then extensive training within the VR should also lead to visual flow response suppression during locomotion even in mice without artificial LC axon stimulation. To

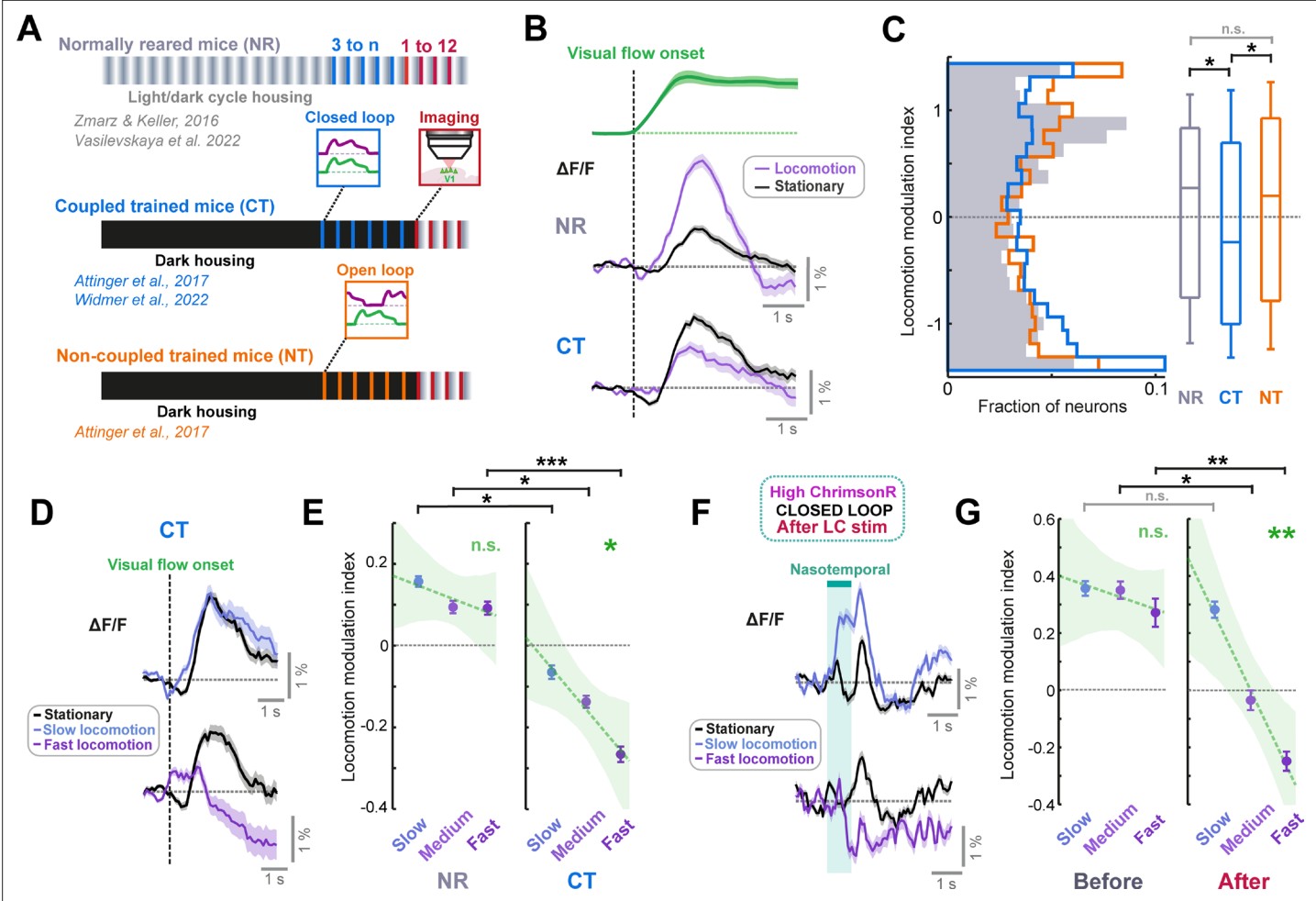

**Figure 5.** LC axon stimulation over minutes recapitulates the visuomotor plasticity seen over days. (**A**) Diagram of the three groups of mice reared with different visuomotor experience. Normally reared mice (NR, gray) were raised with a normal light/dark cycle in their cages, with the full diversity of visuomotor coupling in freely moving conditions. Coupled trained mice (CT, blue) were reared in the dark, and their only visuomotor experience prior to imaging was during closed loop conditions in the virtual reality system. Non-coupled trained mice (NT, orange) were reared similar to coupled trained mice, except their only visual experience was during open loop conditions in the virtual reality system. (**B**) Average population responses of layer 2/3 neurons in V1 to the onset of visual flow during open loop replay sessions. Green shows the average speed profile of the visual flow stimulus, purple shows the average population responses during locomotion, and black shows the average population responses during stationary periods for normally reared (top) and coupled trained (bottom) mice. Shading indicates SEM. (**C**) Distribution of the locomotion modulation indices of all V1 layer 2/3 neurons for the visual flow onset responses recorded in normally reared (gray), coupled trained (blue), and non-coupled trained (orange) mice. Here and in other panels, n.s.: not significant, *: p<0.05, **: p<0.01, ***: p < 0.001. For complete statistical information see *Supplementary file 1*. (**D**) Average population responses of layer 2/3 neurons in V1 in coupled trained mice to the onset of visual flow during stationary periods (black), and during slow (top) and fast (bottom) locomotion (purple). Shading indicates SEM. (**E**) Comparison of locomotion modulation indices averaged across neurons for visual flow onset responses recorded during slow, medium, and fast locomotion speeds, in normally reared (left) and coupled trained (right) mice. Error bars indicate SEM. Green dashed line indicates linear regression fit to the data, and shading indicates the 95% confidence interval of the regression. Green asterisks and 'n.s.' indicate significance of the fit. (**F**) As for panel **D**, but for nasotemporal visual flow responses in high ChrimsonR-expressing mice after LC axon stimulation in closed loop conditions. (**G**) As for panel **E**, but for nasotemporal visual flow responses in high ChrimsonR-expressing mice before (left) and after (right) LC axon stimulation in closed loop visuomotor coupling.

The online version of this article includes the following figure supplement(s) for figure 5:

**Figure supplement 1.** Comparison of locomotion modulation index across different functional neuronal types in layer 2/3 V1.

assess this idea, we reanalyzed two-photon calcium imaging data from V1 layer 2/3 in coupled trained mice reared with visual experience entirely constrained to that of the visuomotor coupling in the VR across days (and otherwise dark reared) (*Attinger et al., 2017*; *Widmer et al., 2022*; *Figure 5A*). We compared this data to similar data from normally reared mice that also have experience of the virtual reality system, but that had experience of normal visuomotor coupling in freely moving conditions

(*Vasilevskaya et al., 2022*; *Zmarz and Keller, 2016*; *Figure 5A*). Analyzing responses to nasotemporal visual flow onsets during open loop replay, we found that visual flow responses were amplified during locomotion relative to stationary periods in normally reared mice but were suppressed during locomotion in coupled trained mice (*Figure 5B*). This resulted in a significantly lower locomotion modulation index in coupled trained mice compared to normally reared mice (*Figure 5C*). The effect was not due to dark rearing, as non-coupled trained mice (also dark reared, but with only open loop virtual reality experience *Figure 5A*; *Attinger et al., 2017*) showed a positive locomotion modulation index on average that did not significantly differ from that of normally reared mice (*Figure 5C*).

Qualitatively, the comparison of locomotion modulation index between normally reared and coupled trained mice (*Figure 5B–C*) was similar to the change in locomotion modulation index in ChrimsonR-expressing mice for nasotemporal visual flow after LC axon stimulation (*Figure 4C*). This suggests that the increase in locomotion driven suppression after LC axon stimulation is caused by a similar form of plasticity to that seen when visuomotor experience is constrained to the virtual reality across days. If this is the case, we should see similarities in the properties of the locomotion driven suppression. First, we looked at the locomotion speed dependence of the suppression. If the negative locomotion modulation index results from learning to suppress visual responses according to a locomotion driven prediction of visual flow, the degree of suppression should depend on locomotion speed. As locomotion speed increased, visual flow responses were increasingly suppressed in coupled trained mice (*Figure 5D*), while this was not evident in normally reared mice (*Figure 5E*). For nasotemporal visual flow responses in ChrimsonR-expressing mice, locomotion modulation index was positive and did not change with locomotion speed at the start of the imaging session, but after stimulation of LC axons in closed loop conditions, locomotion modulation index became increasingly negative as locomotion speed increased (*Figure 5F–G*). Thus, the locomotion driven suppression of visual flow emerging after stimulating LC axons in closed loop conditions was locomotion speed dependent in a similar manner to the locomotion driven suppression seen in coupled trained mice. Note that in high ChrimsonR-expressing mice, suppression of visual flow was seen at high locomotion speeds only (*Figure 5G*), but in coupled trained mice, some level of suppression was evident across all speeds (*Figure 5E*). This could be due to differences in visual flow speed presented: for coupled trained mice, the stimuli analyzed are visual flow onsets during open loop replay; thus, the visual flow speed will cover a similar range as the locomotion speed distribution of the mice. For high ChrimsonR-expressing mice, the stimuli are presented at a fixed visual flow speed that often exceeds the locomotion speed of the mouse. Thus, the fact that suppression only appears at the highest locomotion speed in these mice is consistent with a suppression of visual flow input based on a locomotion speed dependent expectation. In addition to the similarities in locomotion speed dependence of suppression in the two groups, the pattern of change in locomotion modulation index across different functional layer 2/3 cell groups also appeared similar between high ChrimsonR-expressing and coupled trained mice (*Figure 5—figure supplement 1*).

Overall, the pattern of differences in locomotion modulation was remarkably similar for ChrimsonR-expressing (*Figure 5G*, and *Figure 5—figure supplement 1D*) and coupled trained mice (*Figure 5E* and *Figure 5—figure supplement 1B*). This indicates that the functional consequence of plasticity evoked by minutes of LC axon stimulation during visuomotor coupling is similar to that seen after experience is developmentally constrained to visuomotor coupling for days.

## Discussion

The locus coeruleus (LC) has long been hypothesized by computational models to report quantities such as unexpected uncertainty (*Yu and Dayan, 2005*) or high level state-action prediction errors (*Sales et al., 2019*). What these quantities have in common is that they essentially are metrics for the inaccuracy of the brain's internal models. One way to implement such a measure would be to integrate prediction errors from across the brain and then broadcast this signal back across the brain. Indeed, we found that the LC sends unsigned visuomotor prediction errors non-specifically across the dorsal cortex (*Figures 1 and 2*). Models hypothesizing that the LC reports the global inaccuracy of the brain's internal models suggest that the purpose of this is to increase learning rates in output circuits like the cortex, enabling internal models to be modified more rapidly. In line with this, we found that stimulating cortical LC axons during closed loop visuomotor coupling over minutes reduces and even reverses the canonical locomotion related gain of visual flow responses in layer 2/3 of V1, in a manner

specific to the direction of the movement-coupled stimulus (*Figure 4*). Surprisingly, this enhanced plasticity seems to happen even with only a small direct effect of LC axon stimulation on average calcium responses (*Figure 3*). This plasticity, on the timescale of minutes, recapitulates the effect of restricting visuomotor experience to the closed loop coupling in virtual reality across days (*Figure 5*), suggesting that it reflects an acceleration of a slower form of visuomotor plasticity (i.e., an increased learning rate).

The LC responds to unexpected stimuli of various modality, including reward prediction errors, punishment prediction errors, and novel or intense unexpected sensory stimuli (*Aston-Jones and Bloom, 1981*; *Basu et al., 2022*; *Bouret and Sara, 2004*; *Breton-Provencher et al., 2022*; *Deitcher et al., 2019*; *Foote et al., 1980*; *Hervé-Minvielle and Sara, 1995*; *Su and Cohen, 2022*; *Takeuchi et al., 2016*; *Vankov et al., 1995*). The responses we found in LC axons to the various visual flow stimuli were proportional to the change in the degree of error between visual flow and locomotion speeds (*Figure 1*). This supports a model in which the LC signals prediction errors, and combined with previous findings, suggests that prediction errors of all modalities are a general feature of LC signaling. Both positive and negative visuomotor prediction errors appear to be computed in layer 2/3 of the primary visual cortex (*Jordan and Keller, 2020*) and activate the anterior cingulate cortex (*Heindorf and Keller, 2022*) – an area that projects directly to the LC. Thus, the LC likely inherits prediction errors computed upstream in areas like the cortex. Based on the multimodality and unsigned nature of LC responses, and the integrate and broadcast nature of its input-output anatomy (*Schwarz et al., 2015*), our working model of LC function is as follows: The LC integrates prediction errors of all modalities to compute a measure of how inaccurately internal models are currently predicting the global sensory inputs of the animal. The LC does not signal what *type* of prediction error occurred, but what the global rate of prediction errors currently is (i.e. a measure of surprise). The resulting LC output does not directly *drive* model updating in cortical output areas, but it modulates the rate at which local cortical prediction errors drive plasticity in internal models. Thus, the specificity of how internal models are changed depends on ongoing prediction errors in the cortex, and the LC input acts to control the learning rate, modulating the extent to which these cortical prediction errors drive plasticity. This would allow the LC to be a controller of plasticity in internal models: when prediction error rates are high (i.e. during developmental, novel, or volatile situations), updating of internal models will be more rapid as a result of increased LC output. To support this hypothesis, it is important for future work to expand the range of LC stimulation parameters tested to identify the function between LC activation level and cortical plasticity, as well as cortical representations.

While the idea of a brain wide average over sensory prediction errors can explain many of the observed LC responses, other responses are less easily explained by prediction error. LC calcium responses to locomotion onsets, which have been shown previously in LC axons (*Deitcher et al., 2019*; *Reimer et al., 2016*), were generally larger than visual flow or visuomotor mismatch responses (*Figure 2* and *Figure 2—figure supplement 1*). This could be the result of a direct movement related input unrelated to prediction errors, or the result of prediction errors that are not driven by differences between forward locomotion and backward visual flow. Increases in activity occurring after locomotion onset could reflect vestibulomotor prediction errors that result from the fact that the mouse is head-fixed, or visuomotor prediction errors for motor attempts we do not record, like turns or head movements. We do find that visuomotor prediction errors contribute to locomotion onset responses, as locomotion onset responses in the open loop condition are smaller during visual flow, when locomotion evoked visuomotor error is lower (*Figure 2—figure supplement 1B*). However, LC calcium activity also rises prior to locomotion onset, creating the additional possibility of an internally generated movement onset response. Other kinds of movement are also correlated with LC firing, including pup retrievals (*Dvorkin and Shea, 2022*) and lever pressing (*Breton-Provencher et al., 2022*). These movement onset responses could also reflect other forms of prediction error, including action prediction errors (*Greenstreet et al., 2022*) or temporal difference reward prediction errors (in cases where the movement is associated with reward expectation) similar to those documented in the dopaminergic system (*Kim et al., 2020*). Future work will need to investigate these possibilities to build a complete general model of LC computation.

Depleting noradrenaline has been shown to prevent forms of sensory-guided developmental plasticity in the cortex, including ocular dominance plasticity in V1 (*Kasamatsu and Pettigrew, 1976*) and changes in the tonotopic map in the auditory cortex (*Shepard et al., 2015*). Loss of function

experiments can suffer from the potential confound that catecholamines may simply be necessary for normal neuronal activity, and the loss of plasticity may be a secondary effect of a disruption in neuronal function. Indeed, acute noradrenergic receptor block can silence neurons in layer 2/3, and massively reduce the amplitude of visual inputs (*Polack et al., 2013*). In gain-of-function experiments, electrical stimulation of the LC has been shown to partially restore ocular dominance plasticity in adult cats (*Kasamatsu et al., 1985*). Here, we also performed a gain of function experiment: we show that a few minutes of optogenetic LC axon stimulation in V1 of adult mice during visuomotor coupling can recapitulate the developmental effects of constrained exposure to visuomotor coupling in the virtual reality (*Figures 4 and 5*). While our optogenetic stimulation is likely far more constrained to LC neurons than previous electrical stimulations, we cannot rule out a systemic effect of the LC axon stimulation. We know that we are likely antidromically activating the LC with our optogenetic stimulation owing to the resulting pupil dilation (*Figures 3C and 4B*). Given that LC neurons are electrically coupled (*McKinney et al., 2022*) and have widely divergent axons, our stimulation likely has effects on other LC output targets, such as other neuromodulatory systems, and other areas of the cortex. Thus, the plasticity effect that we measure may not entirely result from LC output in V1 alone. However, since the LC will never be acting in isolation and broadcasts its output to different cortical areas, the functional impact of LC axon stimulation in this experiment is strongly suggestive of a role in facilitating plasticity. Future work will be needed to understand the circuit locus of plasticity generated by this paradigm, and its dependence on particular neuromodulators (e.g. dopamine and noradrenaline). The fact that the plasticity occurs despite only minor effects of LC stimulation on neuronal activity (*Figure 3*) indicates that the LC can either enhance plasticity without affecting activity levels, for instance via the molecular cascades or eligibility traces activated by GPCRs (*He et al., 2015*; *Hong et al., 2022*; *Seol et al., 2007*), and/or that a part of the relevant circuit outside of layer 2/3 is more strongly modulated by LC activation.

Locus coeruleus activation has been shown to facilitate learning rates at the behavioral level (*Glennon et al., 2023*; *Glennon et al., 2019*; *Martins and Froemke, 2015*). The behavioral relevance of the LC-gated plasticity in our study is yet to be demonstrated, but we can conjecture some perceptual functions of the plasticity based on its nature: learned suppression of reafferent visual flow during locomotion. This theoretically would be useful at the perceptual level for detection of external moving objects during movement, or for detecting changes in the geometry of the environment through which the animal is moving. Novel behavioral paradigms for assessing these percepts could be used in future studies to demonstrate the behavioral impact of the plasticity that we see.

Prediction-error-driven catecholamine release is thought to gate cortical synaptic plasticity during reinforcement learning (*He et al., 2015*; *Roelfsema and Holtmaat, 2018*). The results presented here expand this idea from reward associations to sensorimotor associations. We propose that a general function of catecholamines is to gate the plasticity underlying predictive learning across all modalities.

## Methods
### Animals and husbandry
All animal procedures were approved by and carried out in accordance with guidelines of the Veterinary Department of the Canton Basel-Stadt, Switzerland, under license number 2573. Mice used in the study were of the NET-Cre strain (*Wagatsuma et al., 2018*) bred in-house, either heterozygous transgenics or (in the case of some controls) wild types. Both males and females were used, and all mice were aged between 8 and 20 weeks. Mice had ad libitum access to water and regular mouse chow throughout the entire study. Mice were co-housed in groups of 2–5 in a reversed 12 hr light-dark cycle, and all imaging experiments took place during the dark part of the cycle.

### Viral vector injections and surgery
For all surgical procedures, mice were anesthetized with an intraperitoneal injection of a mixture of fentanyl (0.05 mg/kg), medetomidine (0.5 mg/kg), and midazolam (5 mg/kg) and provided with both peri- and postoperative general analgesics: Metacam (5 mg/kg, s.c.) and buprenorphine (0.1 mg/kg, s.c.). A lidocaine and ropivacaine mixture was injected subcutaneously into the scalp prior to incision. Metacam analgesia was provided for a further 48 hr post-surgery. Mice were allowed to recover for 1–3 weeks prior to first head-fixation.

### Surgeries for two-photon imaging of LC axons

Thirteen male and female NET-Cre mice (4–6 weeks old) were unilaterally injected at stereotactic coordinates (relative to bregma and brain surface): 5.45 mm posterior, 1.10 mm lateral, 3.65 mm deep. 250–500 nl of AAV2/5-hSyn-DIO-axon-GCaMP6s ($2.7 \times 10^{13}$ genome copies/ml) was injected into each site. A 4 mm diameter circular imaging window was then implanted. A circular craniotomy was made overlying visual cortex (8 mice) or motor cortex (4 mice) and a durectomy was performed. A glass window was then placed onto the craniotomy and fixed in place with cyanoacrylate. A custom titanium headplate was attached to the skull using dental cement (Paladur, Kulzer).

### Surgeries for two-photon imaging in layer 2/3 V1 during optogenetic stimulation of LC axons

Male and female NET-Cre mice (4–6 weeks old) were unilaterally injected with 250–500 nl of AAV2/1-hSyn-DIO-ChrimsonR-tdTomato ($1.2 \times 10^{13}$–$3.4 \times 10^{13}$ genome copies/ml) into the right LC at stereotactic coordinates (relative to bregma and brain surface): 5.45 mm posterior, 1.10 mm lateral, 3.65 mm deep. A maximum of 250 nl of AAV2/1-hSyn-soma-jGCaMP8m (titer range from $10^{13}$–$10^{15}$ genome copies/ml) was then injected across 3–5 locations spanning right V1, and a window and headplate were implanted as above, but with the additional measure of black pigmenting of the dental cement for the reduction of optogenetic stimulation light reaching the eyes of the mouse. For control mice, the injection into the LC was omitted. For the dataset in *Figure 3*, eight mice were injected in the LC, but two were excluded due to lack of expression (see section 'Axon density'). Six mice were not injected in the LC to serve as controls. For the dataset in *Figure 4*, 12 mice were injected in the LC, with 6 mice classified as high ChrimsonR-expressing and 6 mice classified as low ChrimsonR-expressing (see section 'Axon Density'). 7 mice were not injected in the LC to serve as controls.

### Immunohistochemistry

For all experiments with vector injections in LC to express either GCaMP or ChrimsonR-tdTomato, we histologically verified that expression was confined to the neurons within the anatomical location and spatial pattern expected of the LC. In a subset of mice used for imaging LC axons, we also confirmed that the transgene expressing neurons were catecholaminergic by immunostaining for tyrosine hydroxylase and GFP (*Figure 1—figure supplement 1A*). After transcardial perfusion, brains were extracted and placed in 4% paraformaldehyde overnight. Brains were then embedded in 4% agar and sliced at 40 µm on a vibratome. Free floating slices were then washed in PBS-triton, blocked for 2 h with 10% goat serum, and incubated overnight with primary antibodies (1:1000 dilution rabbit anti-tyrosine hydroxylase (Millipore 657012, RRID:AB_10681344), and 1:500 dilution chicken anti-GFP (Abcam ab13970, RRID:AB_300798)). The following day, slices were washed again in PBS-triton and incubated for 2 h with the secondary antibodies at 1:1000 dilution (Alexa Fluor 488 anti-chicken (Jackson ImmunoResearch 703-545-155, RRID:AB_2340375) and Alexa Fluor 568 anti-rabbit (ThermoFisher Scientific, A10042, RRID:AB_2534017)). Slices were mounted in Fluoroshield with DAPI (F6057, Sigma Aldrich).

### Virtual reality and visual stimuli

A virtual tunnel, with walls patterned with vertical sinusoidal gratings, was projected onto a toroidal screen spanning 240 degrees horizontally and 100 degrees vertically of the visual field. The projector output was gated by a 24 kHz TTL such that it only turned on at the turnaround points of the two-photon resonance scanner, minimizing light artifacts during imaging. Mice were head-fixed on an air-floated polystyrene ball of 20 cm diameter. Movement of the virtual tunnel walls could occur only in one dimension (forward and backwards), with the ball restricted to rotation around the pitch axis using a pin. In closed loop conditions, visual flow speed in the tunnel was coupled to the rotation of the ball, with the exception of brief 1 s mismatch stimuli. These were triggered every 12±2 s (mean ± SD) regardless of locomotion behavior and would clamp the visual flow speed to zero. In open loop conditions, the visual flow speed was controlled irrespective of mouse locomotion. Three different types of open loop session were used: (1) for LC axon imaging (*Figures 1 and 2*), as well as optogenetic stimulation of LC axons during open loop replays (*Figure 4*), the visual flow from a previous closed loop session would be replayed. For LC axon imaging, the visual flow onset responses and playback halt responses were analyzed during this session. (2) For stimulation of LC axons to assess

the effect on cortical responses (*Figure 3*), 1 s fixed speed nasotemporal visual flow stimuli would be presented every 9±3 s (mean ± SD). We presented three different speeds of visual flow in a pseudo-random order. (3) For assessing the effects of LC axon stimulation on visuomotor plasticity (*Figure 4*), we presented 1 s duration fixed speed visual flow stimuli every 7±1 s (mean ± SD), but this time only one speed was presented (equivalent to the speed seen during 24 cm/s locomotion in closed loop conditions), however, the direction was either nasotemporal or temporonasal, in a pseudorandom order. All mice imaged were habituated to head-fixation and the virtual reality tunnel for at least four daily 1-hr sessions prior to imaging, until they showed regular, comfortable locomotion.

## Two-photon imaging

Imaging was performed on a modified Thorlabs Bergamo II two-photon microscope system. Excitation illumination was provided by a Ti-Sapphire laser with a wavelength at 930 nm and a laser power between 20 and 30 mW under the objective. When imaging neurons in V1, a piezo scanner was used to move the objective to four separate layers sequentially (15 Hz effective frame rate), but for LC axons, imaging was fixed in a single layer (60 Hz frame rate). Field of view size was 750×400 pixels (approximately 300 x 300 μm). Custom software (available at https://sourceforge.net/projects/iris-scanning; copy archived at *Widmer and Keller, 2023*) was used to acquire the imaging data. For mice undergoing imaging of layer 2/3 V1 neurons (*Figures 3 and 4*), two-photon imaging sites were confirmed as being in V1 by mapping V1 boundaries using intrinsic signal optical imaging.

## Optogenetic stimulation of LC axons

The beam from a 637 nm OBIS laser (Coherent) was focused onto the imaging site via the two-photon microscope objective. The laser was gated to turn on only during the turnaround times of the two-photon resonance scanner, to minimize stimulation induced light artifacts. During stimulation, the laser was presented in 15 ms pulses at 20 Hz (30% duty cycle) for a total duration of 1 s. For stimulation during visual stimulus presentations (*Figure 3*), the power was 27 mW/mm$^2$. Since this evoked a small visual response in the V1 calcium activity of both ChrimsonR-expressing and control mice (*Figure 3C*), the power was reduced for subsequent experiments to 20 mW/mm$^2$ (*Figure 4*), at which point the positive visual response disappeared (*Figure 4—figure supplement 1C*). For the optogenetic stimulation during visual stimulus presentations (*Figure 3*), optogenetic stimulation occurred simultaneous with a random 50% of mismatch stimuli and visual flow stimuli, or in isolation, every 12 s on average, during closed loop conditions. For optogenetic stimulation in the plasticity experiments (*Figure 4*), stimulation occurred every 7 s on average. In 17/61 included imaging sites, this was gated by mouse locomotion such that stimulation only occurred when locomotion speed exceeded 4 cm/s. Since the results looked very similar for both locomotion gated and ungated stimulations, the data were pooled for analysis.

## ΔF/F calculation

Raw images were full-frame registered to correct for lateral brain motion. For V1 layer 2/3 neurons, neuronal somata were manually selected based on mean and maximum fluorescence images. Average fluorescence per region of interest (ROI) was corrected for slow fluorescence drift over time using an 8[th] percentile filter and a 1000 frame window, and divided by the median value over the entire trace to calculate ΔF/F (*Dombeck et al., 2007*). For LC axonal data (*Figures 1 and 2*), there were a number of alternative/additional steps. First, all ROIs within an image above a manually set fluorescence threshold were selected. Non-axonal ROIs from this set were then removed in two ways: first, a 'circularity index' (circularity index = $4\pi$ x area/perimeter$^2$) and area threshold were used. ROIs with circularity index above 0.14 or a total area below 150 pixels were excluded – this served to remove non-axonal structures. Second, any ROI without evidence of calcium activity was removed: a fast Fourier transform was used to obtain the power spectrum of the ΔF/F trace. From this, the signal-to-noise ratio of the calcium signal was calculated, where noise was calculated as the average power between 3 and 8 Hz. ROIs with power below a signal-to-noise ratio of 10 in the frequency band from 0.05 to 1 Hz (classified as signal) were excluded. This served to remove uncommon elongated non-axonal ROIs, such as blood vessel walls. Next, due to the lower signal to noise of the axonal recordings, the ΔF/F recorded at 60 Hz was low pass filtered at 10 Hz. Finally, due to the high correlation of

ΔF/F between LC axonal ROIs within a field of view (*Figure 1—figure supplement 1D–F*), ΔF/F was averaged across all axonal ROIs within a field of view.

## Calculation of visuomotor error (Figure 1)

Visuomotor error is calculated as the difference between locomotion speed and visual flow speed, assuming a constant gain between the two that the mouse predominantly experiences in the virtual reality. The change in absolute error (Δ|Error|) is calculated for each trial as the change in absolute (i.e., unsigned) visuomotor error during the stimulus window, minus the absolute visuomotor error in the 1 s preceding the stimulus. For each type of stimulus and each FoV, Δ|Error| was then averaged across trials. Note that for visuomotor mismatch, Δ|Error| is equivalent to the locomotion speed during mismatch (since visual flow speed during mismatch is zero, and the preceding 1 s is closed loop, where the error is zero). In all stimulus conditions (*Figure 1H*), the Δ|Error| is calculated in a window 0.66 s (40 frames) preceding the window used to calculate ΔF/F responses.

## Locomotion onsets and visual flow onsets

Locomotion onsets were defined as the locomotion speed crossing a threshold of 0.4 cm/s, where the average speed in the previous two seconds was <0.4 cm/s, and where the average speed in the subsequent 1 s exceeded 2 cm/s. The same criteria were used to determine visual flow onsets during open loop replays, using the visual flow speed.

## LC axonal responses

For each visual stimulus trigger, the average ΔF/F in the 1 s window prior to stimulus onset was subtracted from the response, before averaging across trials to get the average response of each axon segment. Mismatch and playback halt responses were calculated for each FoV in the 1 s window beginning 0.66 s (40 frames) after stimulus onset. For visual flow onsets during open loop replays, the window was delayed by 400ms (25 frames) to take into account the slower onset of the stimulus compared to mismatches. Trials were classified as occurring during locomotion if the locomotion speed in the 1 s after the onset of visual flow or mismatch exceeded 1 cm/s, while they were classified as occurring during stationary periods if the locomotion speed was below 1 cm/s. For *Figure 1H*, fast locomotion during mismatch was classified as locomotion speed exceeding 10 cm/s, while slow locomotion was classified as locomotion speed between 1 cm/s and 10 cm/s. For locomotion onset responses (*Figure 2—figure supplement 1*), the average ΔF/F in the 500ms window 2 s prior to loco-motion onset was subtracted from the response, before averaging across trials. Locomotion onsets were considered to take place during the absence of visual flow if average visual flow speed in the 2 s window centered on locomotion onset was below that corresponding to a 1 cm/s locomotion speed in closed loop conditions. If visual flow speed exceeded this value, the trial was considered to take place during visual flow. Responses for all stimuli were averaged for each FoV across trials only if there were at least three valid trials. For all stimulus conditions that were selected based on locomotion speed (e.g. mismatches), 120 sham triggers were generated and sham responses calculated from triggers that were sub-selected based on the same locomotion selection criteria. The average sham response was subtracted from the stimulus response.

## V1 layer 2/3 neuron stimulus responses

For each stimulus trigger, the average ΔF/F in the 1 s window prior to stimulus onset was subtracted from the response, before averaging across trials to get the average response of the neuron. Visual flow and mismatch responses were calculated for each neuron in the window 0.33 s to 2.33 s (5–30 frames) after stimulus onset. Trials were classified as occurring during locomotion if the locomotion speed in the 1 s during visual flow or mismatch exceeded 4 cm/s, while they were classified as during stationary periods if the locomotion speed was below 1 cm/s. For mismatches, only locomotion trials were included. Note that for *Figure 3*, visual responses were not segregated into locomotion and stationary trials: all trials were included. For analysis of locomotion modulation index in different locomotion speed conditions (*Figure 5D–G*), slow, medium, and fast locomotion speed trials were categorized where average locomotion speed during visual flow was between 1 and 5 cm/s, 5 and 10 cm/s, and above 10 cm/s respectively. For the previously published datasets (*Figure 5*, CT, NR, and NT; *Attinger et al., 2017*; *Vasilevskaya et al., 2022*; *Widmer et al., 2022*), differences in the measurement of

locomotion speed across the datasets made it difficult to use a common threshold value. Thus, to make the locomotion modulation indices comparable to those calculated from the newly acquired dataset (*Figure 5D–G*), we used the 95th percentile of the locomotion speed to normalize thresholds across datasets. In addition, since the visual responses used to measure locomotion modulation index were visual flow onsets in open loop conditions rather than rapid onset fixed speed stimuli, the analysis window relative to stimulus onset was shifted by 400ms to account for the slower onset. Responses were averaged for each neuron across trials only if there were at least three valid trials. For all stimulus conditions that were selected based on locomotion speed (e.g. mismatches), the average response to 1000 sham triggers sub-selected based on the same locomotion conditions was calculated. This sham response was subtracted from the stimulus response.

## Locomotion modulation index

To quantify the change in mean visual flow response during locomotion for each neuron normalized by the overall size of response, we calculated a locomotion modulation index (LMI) for each neuron as follows:

$$LMI = \frac{R_{VFloco} - R_{VFstat}}{\sqrt{R_{VFloco}^2 + R_{VFstat}^2}}$$

Where $R_{VFloco}$ is the average visual flow response during locomotion, and $R_{VFstat}$ is the average visual flow response during stationary periods.

## Exclusion criteria in analysis of visual plasticity (Figure 4)

Imaging sessions during which the mouse spent less than 15% of the time locomoting at a speed exceeding 4 cm/s were excluded (13 of 71 sessions), as this would limit both the amount of visuo-motor coupling experienced during the plasticity assay, and our ability to analyze visual responses during locomotion.

## Layer 2/3 functional cell type classification (Figure 3—figure supplement 2)

Neurons were grouped into visually suppressed and mismatch activated NPE neurons, visually activated PPE neurons, and an intermediate group ('other'). Since we were analyzing the visual and mismatch responses of these groups, we used locomotion onset responses to group neurons to avoid a circular analysis. To compute each neuron's locomotion onset response, for each onset the average ΔF/F in the 1 s window beginning 2 s prior to locomotion onset was subtracted from the response, before averaging across trials. Average responses to locomotion onsets (quantified in the window 0.66–4 s [10–60 frames] after onset) in open loop conditions were subtracted from locomotion onset responses in closed loop conditions (where visual flow is concurrent with locomotion) for each neuron. Note that for analyzing the dataset in *Figure 4*, only the open loop locomotion onsets before LC axon stimulation were used. It was previously shown that the difference between locomotion onset responses in open and closed loop conditions correlates with these cell groups (*Jordan and Keller, 2020*). For each dataset, the 33rd and 66th percentile of these values were used as thresholds. Neurons with response differences exceeding the 66th percentile were classified as PPE neurons, and those with response differences below the 33rd percentile were classified as NPE neurons, with the remaining neurons were classified as 'other'. Responses to mismatches and visual flow stimuli were consistent with these groupings in all three datasets (*Figure 3—figure supplement 2*).

## Axon density

PFA fixed brains of mice injected with a vector to express ChrimsonR-tdTomato in NET-positive neurons of the LC were sliced at 40 µm and after mounting, examined under a confocal microscope at ×20 magnification. For all mice, images were taken from four sites in V1, each one from a different cortical slice, with sites chosen based on somatic labelling in the green channel (indicating GCaMP expression and a likely two-photon imaging site). To quantify LC axon density, the red channel images were then analyzed. tdTomato labelled axonal segments were manually traced in ImageJ using the NeuronJ plugin (*Meijering et al., 2004*). The total combined length of the axon segments was then divided by the total area of the cortex in the image to calculate axon density in each image. The average density

across four images was taken as the axon density for each mouse. This value was used to distinguish high expression and low expression mice based on a threshold of 0.002 μm per μm² (total axon length per unit area of the cortex; *Figure 4B* and *Figure 4—figure supplement 1A*). Since only two mice were categorized to have low expression in the dataset used in *Figure 3*, low expression mice were not included as a separate group in these analyses, and instead excluded.

## Pupillometry

Images of the right eye, ipsilateral to the side of LC stimulation, were recorded with a CMOS infrared camera at 30 Hz. Pupil diameter was measured offline by fitting a circle to the pupil, which was backlit by the 930 nm laser light of the two-photon microscope. Pupil diameter traces were z-scored by subtracting the mean of the entire trace and dividing by the standard deviation of the entire trace. To calculate optogenetically induced pupil dilations, the average baseline pupil diameter (in the 1 s prior to stimulus onset) was subtracted from the response for each trial, then the pupil diameter response was averaged across trials. Trials including blinks were excluded.

## Statistical tests

All statistical information for the tests performed in this manuscript are provided in *Supplementary file 1*. For data where the experimental unit was neurons, we used hierarchical bootstrapping (*Saravanan et al., 2020*) for statistical testing due to the nested structure (neurons and mice) of the data. To do this, we first resampled the data (with replacement) at the level of imaging sites. From the selected set of imaging sites, we then resampled the data (with replacement) at the level of neurons. We then computed the mean of this bootstrap sample. This would be repeated 10 000 times to generate a distribution of mean values. For paired tests, the p-value was calculated as the proportion of this distribution that was higher or lower than zero, depending on the null hypothesis. For unpaired tests, the p-value was calculated as the proportion of the distribution higher or lower than the values from the distribution of the compared dataset, depending on the hypothesis. For the bootstrap tests for negative regression slopes (*Figure 5E and G*), the slope of the linear regression fit was calculated for the 10 000 bootstrapped datasets, and the p-value was taken as the proportion of the distribution with a linear regression fit slope exceeding zero. For data where the experimental unit was FoV (*Figures 1 and 2*) or mice (pupil dilation, *Figure 4—figure supplement 1B*), t-tests were used to compare datasets that conformed with normality and did not show statistically distinguishable variances. Datasets that did not conform with normality or had unequal variances were compared with Rank-sum tests. Paired t-tests were used to test the mean of a population against zero when the data was normally distributed, otherwise signed-rank tests were used to test the median against zero (*Figure 1*).

## Acknowledgements

We thank all members of the Keller lab for discussion and support, Tingjia Lu for the production of viral vectors, Susumu Tonegawa for sharing NET-Cre mice, and Andreas Lüthi for reagents and equipment. This project has received funding from Human Frontier Science Program (LT000077/2019 L, long term fellowship to RJ), the Swiss National Science Foundation, the Novartis Research Foundation, and the European Research Council (ERC) under the European Union's Horizon 2020 research and innovation programme (grant agreement no. 865617).

## Additional information

### Funding

| Funder | Grant reference number | Author |
| --- | --- | --- |
| Human Frontier Science Program | LT000077/2019-L | Rebecca Jordan |
| H2020 European Research Council | 865617 | Georg B Keller |

| Funder | Grant reference number | Author |
|---|---|---|
| Novartis Research Foundation | | Georg B Keller |
| Schweizerischer Nationalfonds zur Förderung der Wissenschaftlichen Forschung | | Georg B Keller |

The funders had no role in study design, data collection and interpretation, or the decision to submit the work for publication.

## Author contributions

Rebecca Jordan, Conceptualization, Data curation, Software, Formal analysis, Funding acquisition, Investigation, Visualization, Methodology, Writing – original draft, Writing – review and editing; Georg B Keller, Resources, Supervision, Funding acquisition, Methodology, Writing – original draft, Project administration, Writing – review and editing

## Author ORCIDs

Rebecca Jordan http://orcid.org/0000-0002-4871-6265
Georg B Keller http://orcid.org/0000-0002-1401-0117

## Ethics

All animal procedures were approved by and carried out in accordance with guidelines of the Veterinary Department of the Canton Basel-Stadt, Switzerland, under license number 2573.

Reviewer #1 (Public Review): https://doi.org/10.7554/eLife.85111.3.sa1
Reviewer #2 (Public Review): https://doi.org/10.7554/eLife.85111.3.sa2
Author Response https://doi.org/10.7554/eLife.85111.3.sa3

# Additional files

## Supplementary files

- Supplementary file 1. Table detailing all statistical tests and sample sizes.
- MDAR checklist

## Data availability

All data and analysis code are available at https://data.fmi.ch/PublicationSupplementRepo/, in the section for the group of Georg Keller, under the title 'The locus coeruleus broadcasts prediction errors across the cortex to promote sensorimotor plasticity'. Other previously published datasets used in this study (*Figure 5*) are located in the same repository, listed under the same titles as the published papers in which they were first used (*Attinger et al., 2017*; *Vasilevskaya et al., 2022*; *Widmer et al., 2022*; *Zmarz and Keller, 2016*). Data and analysis code are also available at Zenodo under https://doi.org/10.5281/zenodo.8006972. Core analysis and imaging code are available at https://sourceforge.net/projects/iris-scanning/ (copy archived at *Widmer and Keller, 2023*).

The following dataset was generated:

| Author(s) | Year | Dataset title | Dataset URL | Database and Identifier |
|---|---|---|---|---|
| Jordan R, Keller GB | 2023 | The locus coeruleus broadcasts prediction errors across the cortex to promote sensorimotor plasticity | https://doi.org/10.5281/zenodo.8006972 | Zenodo, 10.5281/zenodo.8006972 |

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
