## [Editor Report · eLife assessment]

This **important** study provides **convincing** evidence that locus coeruleus is activated during visuomotor mismatches. Gain of function optogenetic experiments complement this evidence and indicate that locus coeruleus could be involved in the learning process that enables visuomotor predictions. This study, therefore, sets the groundwork for the circuit dissection of predictive signals in the visual cortex. Loss-of-function experiments would strengthen the evidence of the involvement of locus coeruleus in prediction learning. These results will be of interest to systems neuroscientists.

---

## [Referee Report · Reviewer #1 (Public Review)]

Jordan and Keller investigated the possibility that sensorimotor prediction error (mismatch between expected and actual inputs) triggers locus coeruleus (LC) activation, which in turn drives plasticity of cortical neurons that detect the mismatch (e.g. layer 2/3 neurons in V1), thus updating the internal presentation (expected) to match more the sensory input. Using genetic tools to selectively label LC neurons in mice and in vivo imaging of LC axonal calcium responses in the V1 and motor cortex in awake mice in virtual reality training, they showed that LC axons responded selectively to a mismatch between the visual input and locomotion. The greater the mismatch (the faster the locomotion in relation to the visual input), the larger the LC response. This seemed to be a global response as LC responses were indistinguishable between sensory and motor cortical areas. They further showed that LC drove learning (updating the internal model) despite that LC optical stimulation failed to alter acute cellular responses. Responses in the visual cortex increased with locomotion, and this was suppressed following LC phasic stimulation during visuomotor coupled training (closed loop). In the last section, they showed that artificial optogenetic stimulation of LC permitted plasticity over minutes, which would normally take days in non-stimulated mice trained in the visuomotor coupling mode. These data enhance our understanding of LC functionality in vivo and support the framework that LC acts as a prediction error detector and supervises cortical plasticity to update internal representations.

The experiments are well-designed and carefully conducted. The conclusions of this work are in general well supported by the data.

---

## [Referee Report · Reviewer #2 (Public Review)]

The work presented by Jordan and Keller aims at understanding the role of noradrenergic neuromodulation in the cortex of mice exploring a visual virtual environment. The authors hypothesized that norepinephrine released by Locus Coeruleus (LC) neurons in cortical circuits gates the plasticity of internal models following visuomotor prediction errors. To test this hypothesis, they devised clever experiments that allowed them to manipulate visual flow with respect to locomotion to create prediction errors in visuomotor coupling and measure the related signals in LC axons innervating the cortex using two-photon calcium imaging. They observed calcium responses proportional to absolute prediction errors that were non-specifically broadcast across the dorsal cortex. To understand how these signals contribute to computations performed by V1 neurons in layers 2/3, the authors activated LC noradrenergic inputs using optogenetic stimulations while imaging calcium responses in cortical neurons. Although LC activation had little impact on evoked activity related to visuomotor prediction errors, the authors observed changes in the effect of locomotion on visually evoked activity after repeated LC axons activation that were absent in control mice. Using a clever paradigm where the locomotion modulation index was measured in the same neurons before and after optogenetic manipulations, they confirmed that this plasticity depended on the density of LC axons activated, the visual flow associated with running, and the concurrent visuomotor coupling during LC activation. Based on similar locomotion modulation index dependency on speed observed in mice that develop only with visuomotor experience in the virtual environment, the authors concluded that changes in locomotion modulation index are the result of experience-dependent plasticity occurring at a much faster rate during LC axons optogenetic stimulations.

The study provides very compelling data on a timely and fascinating topic in neuroscience. The authors carefully designed experiments and corresponding controls to exclude any confounding factors in the interpretation of neuronal activity in LC axons and cortical neurons. The quality of the data and the rigor of the analysis are important strengths of the study. I believe this study will have an important contribution to the field of system neuroscience by shedding new light on the role of a key neuromodulator. The results provide strong support for the claims of the study.

---

## [Author Response]

The following is the authors' response to the original reviews.

**Reviewer #1 (Public Review):**
[…] The experiments are well-designed and carefully conducted. The conclusions of this work are in general well supported by the data. There are a couple of points that need to be addressed or tested.It is unclear how LC phasic stimulation used in this study gates cortical plasticity without altering cellular responses (at least at the calcium imaging level). As the authors mentioned that Polack et al 2013 showed a significant effect of NE blockers in membrane potential and firing rate in V1 layer2/3 neurons during locomotion, it would be useful to test the effect of LC silencing (coupled to mismatch training) on both cellular response and cortical plasticity or applying NE antagonists in V1 in addition to LC optical stimulation. The latter experiment will also address which neuromodulator mediates plasticity, given that LC could co-release other modulators such as dopamine (Takeuchi et al. 2016 and Kempadoo et al. 2016). LC silencing experiment would establish a causal effect more convincingly than the activation experiment.

Regarding the question of how phasic stimulation could alter plasticity without affecting the response sizes or activity in general, we believe there are possibilities supported by previous literature. It has been shown that catecholamines can gate plasticity by acting on eligibility traces at synapses (He et al., 2015; Hong et al., 2022). In addition, all catecholamine receptors are metabotropic and influence intracellular signaling cascades, e.g., via adenylyl cyclase and phospholipases. Catecholamines can gate LTP and LTD via these signaling pathways in vitro (Seol et al., 2007). Both of these influences on plasticity at the molecular level do not necessitate or predict an effect on calcium activity levels. We have now expanded on this in the discussion of the revised manuscript.

While a loss of function experiment could add additional corroborating evidence that LC output is required for the plasticity seen, we did not perform loss-of-function experiments for three reasons:

The effects of artificial activity changes around physiological set point are likely not linear for increases and decreases. The problem with a loss of function experiment here is that neuromodulators like noradrenaline affect general aspects of neuronal function. This is apparent in Polack et al., 2013: during the pharmacological blocking experiment, the membrane hyperpolarizes, membrane variance becomes very low, and the cells are effectively silenced (Figure 7 of (Polack et al., 2013)), demonstrating an immediate impact on neuronal function when noradrenaline receptor activation is presumably taken below physiological/waking levels. In light of this, if we reduce LC output/noradrenergic receptor activation and find that plasticity is prevented, this could be the result of a direct influence on the plasticity process, or, the result of a disruption of another aspect of neuronal function, like synaptic transmission or spiking. We would therefore challenge the reviewer’s statement that a loss-of-function experiment would establish a causal effect more convincingly than the gain- of-function experiment that we performed.The loss-of-function experiment is technically more difficult both in implementation and interpretation. Control mice show no sign of plasticity in locomotion modulation index (LMI) on the 10-minute timescale (Figure 4J), thus we would not expect to see any effect when blocking plasticity in this experiment. We would need to use dark-rearing and coupled- training of mice in the VR across development to elicit the relevant plasticity ((Attinger et al., 2017); manuscript Figure 5). We would then need to silence LC activity across days of VR experience to prevent the expected physiological levels of plasticity. Applying NE antagonists in V1 over the entire period of development seems very difficult. This would leave optogenetically silencing axons locally, which in addition to the problems of doing this acutely (Mahn et al., 2016; Raimondo et al., 2012), has not been demonstrated to work chronically over the duration of weeks. Thus, a negative result in this experiment will be difficult to interpret, and likely uninformative: We will not be able to distinguish whether the experimental approach did not work, or whether local LC silencing does nothing to plasticity. Note that pharmacologically blocking noradrenaline receptors during LC stimulation in the plasticity experiment is also particularly challenging: they would need to be blocked throughout the entire 15 minute duration of the experiment with no changes in concentration of antagonist between the ‘before’ and ‘after’ phases, since the block itself is likely to affect the response size, as seen in Polack et al., 2013, creating a confound for plasticity-related changes in response size. Thus, we make no claim about which particular neuromodulator released by the LC is causing the plasticity.

Note that pharmacologically blocking noradrenaline receptors during LC stimulation in the plasticity experiment is also particularly challenging: they would need to be blocked throughout the entire 15 minute duration of the experiment with no changes in concentration of antagonist between the ‘before’ and ‘after’ phases, since the block itself is likely to affect the response size, as seen in Polack et al., 2013, creating a confound for plasticity-related changes in response size. Thus, we make no claim about which particular neuromodulator released by the LC is causing the plasticity.

3. There are several loss-of-function experiments reported in the literature using different developmental plasticity paradigms alongside pharmacological or genetic knockout approaches. These experiments show that chronic suppression of noradrenergic receptor activity prevents ocular dominance plasticity and auditory plasticity (Kasamatsu and Pettigrew, 1976; Shepard et al., 2015). Almost absent from the literature, however, are convincing gain-of-function plasticity experiments.

Overall, we feel that loss-of-function experiments may be a possible direction for future work but, given the technical difficulty and – in our opinion – limited benefit that these experiments, would provide in light of the evidence already provided for the claims we make, we have chosen not to perform these experiments at this time. Note that we already discuss some of the problems with loss- of-function experiments in the discussion.

2) The cortical responses to NE often exhibit an inverted U-curve, with higher or lower doses of NE showing more inhibitory effects. It is unclear how responses induced by optical LC stimulation compare or interact with the physiological activation of the LC during the mismatch. Since the authors only used one frequency stimulation pattern, some discussion or additional tests with a frequency range would be helpful.

This is correct, we do not know how the artificial activation of LC axons relates to physiological activation, e.g. under mismatch. The stimulation strength is intrinsically consistent in our study in the sense that the stimulation level to test for changes in neuronal activity is similar to that used to probe for plasticity effects. We suspect that the artificial activation results in much stronger LC activity than seen during mismatch responses, given that no sign of the plasticity in LMI seen in high ChrimsonR occurs in low ChrimsonR or control mice (Figure 4J). Note, that our conclusions do not rely on the assumption that the stimulation is matched to physiological levels of activation during the visuomotor mismatches that we assayed. The hypothesis that we put forward is that increasing levels of activation of the LC (reflecting increasing rates or amplitude of prediction errors across the brain) will result in increased levels of plasticity. We know that LC axons can reach levels of activity far higher than that seen during visuomotor mismatches, for instance during air puff responses, which constitute a form of positive prediction error (unexpected tactile input) (Figures 2C and S1C). The visuomotor mismatches used in this study were only used to demonstrate that LC activity is consistent with prediction error signaling. We have now expanded on these points in the discussion as suggested.

**Reviewer #1 (Recommendations For The Authors):**
1) In Figure 3E, there is a rebound response of ChrimsonR at the offset of the mismatch. Is that common? If so, what does it mean? If not, maybe replace it with a more common example trace.

This trace in fact represents the population average, so this offset response (or ‘rebound’) reflects a significant component of the population response to visual flow onset (i.e., mismatch offset), only under conditions of LC stimulation. See our response to reviewer 2 concerning this element of the response.

2) It would be helpful to have some discussions on how a mismatch signal reaches and activates LC from cortical neurons.

We have now added a short segment on this to the discussion.

**Reviewer #2 (Public Review):**
[…] The study provides very compelling data on a timely and fascinating topic in neuroscience. The authors carefully designed experiments and corresponding controls to exclude any confounding factors in the interpretation of neuronal activity in LC axons and cortical neurons. The quality of the data and the rigor of the analysis are important strengths of the study. I believe this study will have an important contribution to the field of system neuroscience by shedding new light on the role of a key neuromodulator. The results provide strong support for the claims of the study. However, I also believe that some results could have been strengthened by providing additional analyses and experimental controls. These points are discussed below.Calcium signals in LC axons tend to respond with pupil dilation, air puffs, and locomotion as the authors reported. A more quantitative analysis such as a GLM model could help understand the relative contribution (and temporal relationship) of these variables in explaining calcium signals. This could also help compare signals obtained in the sensory and motor cortical domains. Indeed, the comparison in Figure 2 seems a bit incomplete since only "posterior versus anterior" comparisons have been performed and not within-group comparisons. I believe it is hard to properly assess differences or similarities between calcium signal amplitude measured in different mice and cranial windows as they are subject to important variability (caused by different levels of viral expression for instance). The authors should at the very least provide a full statistical comparison between/within groups through a GLM model that would provide a more systematic quantification.

To provide a more detailed comparison of responses, we have expanded on the analysis in Figure 2 to include comparative heatmaps from anterior and posterior imaging sites, as well as statistical comparisons of the response curves as a function of time. This shows how similar the responses are in the two regions.

Beyond this, we are not sure how a regression analysis (GLM or otherwise) would help support the main point we aim to make here. The responses in anterior and posterior regions are similar, which supports a broadcast model of LC function in the cortex, rather than specialized routing of prediction error signals to cortical areas. Linear contributions of the signals are apparent from the stimulus triggered responses, and while non-linear interactions between the different variables are certainly an interesting question, they go beyond the point we aim to make and would also not be captured by a regression analysis. In addition, we have refined our language replacing descriptors of ‘the same’ or ‘indistinguishable’ between the two regions with ‘similar’, to highlight that while we find no evidence of a difference, our analysis does not cover all possible differences that might appear when looking at non-linear interactions.

Previous studies using stimulations of the locus coeruleus or local iontophoresis of norepinephrine in sensory cortices have shown robust responses modulations (see McBurney-Lin et al., 2019, https://doi.org/10.1016/j.neubiorev.2019.06.009 for a review). The weak modulations observed in this study seem at odds with these reports. Given that the density of ChrimsonR-expressing axons varies across mice and that there are no direct measurements of their activation (besides pupil dilation), it is difficult to appreciate how they impact the local network. How does the density of ChrimsonR-expressing axons compare to the actual density of LC axons in V1? The authors could further discuss this point.

In terms of estimating the percentage of cortical axons labelled based on our axon density measurements: we refer to cortical LC axonal immunostaining in the literature to make this comparison.

In motor cortex, an average axon density of 0.07 µm/µm2 has been reported (Yin et al., 2021), and 0.09 µm/µm2 in prefrontal cortex (Sakakibara et al., 2021). Density of LC axons varies by cortical area, with higher density in motor cortex and medial areas than sensory areas (Agster et al., 2013): V1 axon density is roughly 70% of that in cingulate cortex (adjacent to motor and prefrontal cortices) (Nomura et al., 2014). So, we approximate a maximum average axon density in V1 of approximately 0.056 µm/µm2.

Because these published measurements were made from images taken of tissue volumes with larger z-depth (~ 10 µm) than our reported measurements (~ 1 µm), they appear much larger than the ranges reported in our manuscript (0.002 to 0.007 µm/µm2). We repeated the measurements in our data using images of volumes with 10 µm z-depth, and find that the percentage axons labelled in our study in high ChrimsonR-expressing mice ranges between 0.012 to 0.039 µm/µm2. This corresponds to between 20% to 70% of the density we would expect based on previous work. Note that this is a potentially significant underestimate, and therefore should be used as a lower bound: analyses in the literature use images from immunostaining, where the signal to background ratio is very high. In contrast, we did not transcardially perfuse our mice leading to significant background (especially in the pia/L1, where axon density is high - (Agster et al., 2013; Nomura et al., 2014)), and the intensity of the tdTomato is not especially high. We therefore are likely missing some narrow, dim, and superficial fibers in our analysis.

We also can quantify how our variance in axonal labelling affects our results: For the dataset in Figure 3, there doesn’t appear to be any correlation between the level of expression and the effect of stimulating the axons on the mismatch or visual flow responses for each animal (Author response image 1), while there is a significant correlation between the level of expression and the pupil dilation, consistent with the dataset shown in Figure 4. Thus, even in the most highly expressing mice, there is no clear effect on average response size at the level of the population. We have added these correlations to the revised manuscript as a new Figure S3.

**Author response image 1. sa3fig1:** Correlations between axon density and average effect of laser stimulation on stimulus responses and pupil dilation (data from manuscript Figure 3). Grey points show control mice, blue points show low ChrimsonR-expressing mice, and purple points show high ChrimsonR- expressing mice.

To our knowledge, there has not yet been any similar experiment reported utilizing local LC axonal optogenetic stimulation while recording cortical responses, so when comparing our results to those in the literature, there are several important methodological differences to keep in mind. The vast majority of the work demonstrating an effect of LC output/noradrenaline on responses in the cortex has been done using unit recordings, and while results are mixed, these have most often demonstrated a suppressive effect on spontaneous and/or evoked activity in the cortex (McBurney- Lin et al., 2019). In contrast to these studies, we do not see a major effect of LC stimulation either on baseline or evoked calcium activity (Figure 3), and, if anything, we see a minor potentiation of transient visual flow onset responses (see also Author Response Figure 2). There could be several reasons why our stimulation does not have the same effect as these older studies:

Recording location: Unit recordings are often very biased toward highly active neurons (Margrie et al., 2002) and deeper layers of the cortex, while we are imaging from layer 2/3 – a layer notorious for sparse activity. In one of the few papers to record from superficial layers, it was been demonstrated that deeper layers in V1 are affected differently by LC stimulation methods compared to more superficial ones (Sato et al., 1989), with suppression more common in superficial layers. Thus, some differences between our results and those in the majority of the literature could simply be due to recording depth and the sampling bias of unit recordings.Stimulation method: Most previous studies have manipulated LC output/noradrenaline levels by either iontophoretically applying noradrenergic receptor agonists, or by electrically stimulating the LC. Arguably, even though our optogenetic stimulation is still artificial, it represents a more physiologically relevant activation compared to iontophoresis, since the LC releases a number of neuromodulators including dopamine, and these will be released in a more physiological manner in the spatial domain and in terms of neuromodulator concentration. Electrical stimulation of the LC as used by previous studies differs from our optogenetic method in that LC axons will be stimulated across much wider regions of the brain (affecting both the cortex and many of its inputs), and it is not clear whether the cause of cortical response changes is in cortex or subcortical. In addition, electrical LC stimulation is not cell type specific.Temporal features of stimulation: Few previous studies had the same level of temporal control over manipulating LC output that we had using optogenetics. Given that electrical stimulation generates electrical artifacts, coincident stimulation during the stimulus was not used in previous studies. Instead, the LC is often repeatedly or tonically stimulated, sometimes for many seconds, prior to the stimulus being presented. Iontophoresis also does not have the same temporal specificity and will lead to tonically raised receptor activity over a time course determined by washout times.State specificity: Most previous studies have been performed under anesthesia – which is known to impact noradrenaline levels and LC activity (Müller et al., 2011). Thus, the acute effects of LC stimulation are likely not comparable between anesthesia and in the awake animal.

Due to these differences, it is hard to infer why our results differ compared to other papers. The study with the most similar methodology to ours is (Vazey et al., 2018), which used optogenetic stimulation directly into the mouse LC while recording spiking in deep layers of the somatosensory cortex with extracellular electrodes. Like us, they found that phasic optogenetic stimulation alone did not alter baseline spiking activity (Figure 2F of Vazey et al., 2018), and they found that in layers 5 and 6, short latency transient responses to foot touch were potentiated and recruited by simultaneous LC stimulation. While this finding appears more overt than the small modulations we see, it is qualitatively not so dissimilar from our finding that transient responses appear to be slightly potentiated when visual flow begins (Author response image 2). Differences in the degree of the effect may be due to differences in the layers recorded, the proportion of the LC recruited, or the fact anesthesia was used in Vazey et al., 2018.

Note that we only used one set of stimulation parameters for optogenetic stimulation, and it is always possible that using different parameters would result in different effects. We have now added a discussion on the topic to the revised manuscript.

In the analysis performed in Figure 3, it seems that red light stimulations used to drive ChrimsonR also have an indirect impact on V1 neurons through the retina. Indeed, figure 3D shows a similar response profile for ChrimsonR and control with calcium signals increasing at laser onset (ON response) and offset (OFF response). With that in mind, it is hard to interpret the results shown in Figure 3E-F without seeing the average calcium time course for Control mice. Are the responses following visual flow caused by LC activation or additional visual inputs? The authors should provide additional information to clarify this result.

This is a good point. When we plot the average difference between the stimulus response alone and the optogenetic stimulation + stimulus response, we do indeed find that there is a transient increase in response at the visual flow onset (and the offset of mismatch, which is where visual flow resumes), and this is only seen in ChrimsonR-expressing mice (Author response image 2). We therefore believe that these enhanced transients at visual flow onset could be due to the effect of ChrimsonR stimulation, and indeed previous studies have shown that LC stimulation can reduce the onset latency and latency jitter of afferent-evoked activity (Devilbiss and Waterhouse, 2004; Lecas, 2004), an effect which could mediate the differences we see. We have added this analysis to the revised manuscript in Figure 3 and added discussion accordingly.

**Author response image 2. sa3fig2:** Difference in responses to visual stimuli caused by optogenetic stimulation, calculated by subtracting the average response when no laser was presented from the average response when the laser was presented concurrent with the visual stimulus. Pink traces show the response difference for ChrimsonR-expressing mice, and grey shows the same for control mice. Black blocks below indicate consecutive timepoints after stimulation showing a significant difference between ChrimsonR and control as determined by hierarchical bootstrapping (p<0.05).

Some aspects of the described plasticity process remained unanswered. It is not clear over which time scale the locomotion modulation index changes and how many optogenetic stimulations are necessary or sufficient to saturate this index. Some of these questions could be addressed with the dataset of Figure 3 by measuring this index over different epochs of the imaging session (from early to late) to estimate the dynamics of the ongoing plasticity process (in comparison to control mice). Also, is there any behavioural consequence of plasticity/update of functional representation in V1? If plasticity gated by repeated LC activations reproduced visuomotor responses observed in mice that were exposed to visual stimulation only in the virtual environment, then I would expect to see a change in the locomotion behaviour (such as a change in speed distribution) as a result of the repeated LC stimulation. This would provide more compelling evidence for changes in internal models for visuomotor coupling in relation to its behavioural relevance. An experiment that could confirm the existence of the LC-gated learning process would be to change the gain of the visuomotor coupling and see if mice adapt faster with LC optogenetic activation compared to control mice with no ChrimsonR expression. Authors should discuss how they imagine the behavioural manifestation of this artificially-induced learning process in V1.

Regarding the question of plasticity time course: Unfortunately, owing to the paradigm used in Figure 3, the time course of the plasticity will not be quantifiable from this experiment. This is because in the first 10 minutes, the mouse is in closed loop visuomotor VR experience, undergoing optogenetic stimulation (this is the time period in which we record mismatches). We then shift to the open loop session to quantify the effect of optogenetic stimulation on visual flow responses. Since the plasticity is presumably happening during the closed loop phase, and we have no read-out of the plasticity during this phase (we do not have uncoupled visual flow onsets to quantify LMI in closed loop), it is not possible to track the plasticity over time.

Regarding the behavioral relevance of the plasticity: The type of plasticity we describe here is consistent with predictive, visuomotor plasticity in the form of a learned suppression of responses to self-generated visual feedback during movement. Intuitive purposes of this type of plasticity would be (1) to enable better detection of external moving objects by suppressing the predictable (and therefore redundant) self-generated visual motion and (2) to better detect changes in the geometry of the world (near objects have a larger visuomotor gain that far objects). In our paradigm, we have no intuitive read-out of the mouse’s perception of these things, and it is not clear to us that they would be reflected in locomotion speed, which does not differ between groups (manuscript Figure S5). Instead, we would need to turn to other paradigms for a clear behavioral read-out of predictive forms of sensorimotor learning: for instance, sensorimotor learning paradigms in the VR (such as those used in (Heindorf et al., 2018; Leinweber et al., 2017)), or novel paradigms that reinforce the mouse for detecting changes in the gain of the VR, or moving objects in the VR, using LC stimulation during the learning phase to assess if this improves acquisition. This is certainly a direction for future work. In the case of a positive effect, however, the link between the precise form of plasticity we quantify in this manuscript and the effect on the behavior would remain indirect, so we see this as beyond the scope of the manuscript. We have added a discussion on this topic to the revised manuscript.

Finally, control mice used as a comparison to mice expressing ChrimsonR in Figure 3 were not injected with a control viral vector expressing a fluorescent protein alone. Although it is unlikely that the procedure of injection could cause the results observed, it would have been a better control for the interpretation of the results.

We agree that this indeed would have been a better control. However, we believe that this is fortunately not a major problem for the interpretation of our results for two reasons:

The control and ChrimsonR expressing mice do not show major differences in the effect of optogenetic LC stimulation at the level of the calcium responses for all results in Figure 3, with the exception of the locomotion modulation indices (Figure 3I). Therefore, in terms of response size, there is no major effect compared to control animals that could be caused by the injection procedure, apart from marginally increased transient responses to visual flow onset – and, as the reviewer notes, it is difficult to see how the injection procedure would cause this effect.The effect on locomotion modulation index (Figure 3I) was replicated with another set of mice in Figure 4C, for which we did have a form of injected control (‘Low ChrimsonR’), which did not show the same plasticity in locomotion modulation index (Figure 4E). We therefore know that at least the injection itself is not resulting in the plasticity effect seen.

**Reviewer #2 (Recommendations For The Authors):**
In experiments where axonal imaging was performed on LC axons, the authors should indicate the number of mice used in addition to the number of Field of View (FoV). Indeed, samples (FoVs) are not guaranteed to be independent as LC axons can span large cortical areas and the same axon can end up in different FoVs. Please provide statistics across mice/cranial windows to confirm the robustness of the results.

All information requested regarding animal numbers in axonal imaging are provided in the statistical Table S1, as well as in the text and figures (e.g., Figure 2A). Samples will be independent in time (as different FoVs were imaged on different days), but it is indeed possible that axon segments from different FoVs within an animal come from the same axon.

Averaging across animals greatly reduces statistical power. We have therefore implemented hierarchical bootstrapping instead: bootstrapping first occurs at the level of animal and then at the level of FoV. All p-values that were reported as significant in manuscript remained significant with this test, with no major reduction in significance level, with the exception of Figure S2B, where statistical significance was lost (p = 0.04 with Rank sum, p = 0.07 with hierarchical Bootstrapping). We therefore conclude that sampling from the same animals across days is not responsible for the significance of results reported.

**Author response table 1. sa3table1:** 

Figure panel	Comparison	Value compared	Mean 1 (delta shown for paired tests)	DS1	Mean 2 (n/a for paired tests)	SD 2	P value	Test type	N (FoVs)	N (Mice)
**Figure 1D**	Mismatch vs Playback halt (stationary)	Mean response [% ΔF/F]	2.25	2.28	-1.08	1.71	<10-4	Bootstrap	40, 25	13, 10
**Figure 1D**	Mean response vs zero	Mismatch response [% ΔF/F]	2.25	2.28	n/a	n/a	<10-4	Bootstrap	40	13
**Figure 1D**	Playback halt locomotion vs stationary	Mean response [% ΔF/F]	-1.08	1.71	0.58	3.54	0.0348	Bootstrap	25, 24	10, 10
**Figure 1D**	Mean response vs zero	Playback halt response (stationary) [% ΔF/F]	-1.08	1.71	n/a	n/a	0.0072	Bootstrap	25	10
**Figure 1D**	Mismatch vs Playback halt response (locomotion)	Mean response [% ΔF/F]	0.58	3.54	2.25	2.28	0.0286	Bootstrap	24, 40	10, 13
**Figure 1D**	Mean response vs zero	Playback halt response (locomotion) [% ΔF/F]	0.58	3.54	n/a	n/a	0.2314	Bootstrap	24	10
**Figure 1G**	Mean response vs zero	Visual flow response (locomotion) [% ΔF/F]	-0.58	4.19	n/a	n/a	0.2821	Bootstrap	21	11
**Figure 1G**	Mean response vs zero	Visual flow response (stationary) [% ΔF/F]	3.16	5.27	n/a	n/a	0.0055	Bootstrap	28	11
**Figure 1G**	Locomotion vs stationary	Mean visual response [% ΔF/F]	3.16	5.27	-0.58	4.19	0.0082	Bootstrap	28, 21	11, 11
**Figure 2B**	Posterior vs Anterior	Mismatch response [% ΔF/F]	2.55	2.02	1.21	2.92	0.0922	Bootstrap	31, 9	9, 4
**Figure 2B**	Posterior vs Anterior	Playback halt response (stationary) [% ΔF/F]	-1.11	1.56	-1.01	2.05	0.499	Bootstrap	16, 9	6,4
**Figure 2B**	Posterior vs Anterior	Visual flow response (stationary) [% ΔF/F]	2.98	5.46	3.53	5.14	0.4194	Bootstrap	19, 9	7, 4
**Figure 2C**	Posterior vs Anterior	Air puff response [% ΔF/F]	24.73	10.03	24.48	11.24	0.4841	Bootstrap	13, 8	6, 4
**Figure 2C**	Posterior vs Anterior	Locomotion response (closed loop) [% ΔF/F]	14.99	7.07	13.33	7.86	0.3666	Bootstrap	28, 9	9, 4
**Figure 2C**	Posterior vs Anterior	Locomotion response (no visual flow) [% ΔF/F]	14.42	8.05	14.96	11.93	0.4204	Bootstrap	8, 6	6, 3
**Figure 2C**	Posterior vs Anterior	Locomotion response (visual flow) [% ΔF/F]	10.85	5.89	8.12	8.46	0.3324	Bootstrap	14, 6	6, 3
**Figure S2B**	With vs without visual flow	Locomotion onset response (open loop) [% ΔF/F]	3.03	7.82	7.17	6.45	0.071	Bootstrap	20, 14	9, 9

Author References:

Agster, K.L., Mejias-Aponte, C.A., Clark, B.D., Waterhouse, B.D., 2013. Evidence for a regional specificityi n the density and distribution of noradrenergic varicosities in rat cortex. Journal of Comparative Neurology 521, 2195–2207. https://doi.org/10.1002/cne.23270Attinger, A., Wang, B., Keller, G.B., 2017. Visuomotor Coupling Shapes the Functional Development of Mouse Visual Cortex. Cell 169, 1291-1302.e14. https://doi.org/10.1016/j.cell.2017.05.023Devilbiss, D.M., Waterhouse, B.D., 2004. The Effects of Tonic Locus Ceruleus Output on Sensory-Evoked Responses of Ventral Posterior Medial Thalamic and Barrel Field Cortical Neurons in the Awake Rat. J. Neurosci. 24, 10773–10785. https://doi.org/10.1523/JNEUROSCI.1573-04.2004He, K., Huertas, M., Hong, S.Z., Tie, X., Hell, J.W., Shouval, H., Kirkwood, A., 2015. Distinct Eligibility Traces for LTP and LTD in Cortical Synapses. Neuron 88, 528–538. https://doi.org/10.1016/j.neuron.2015.09.037Heindorf, M., Arber, S., Keller, G.B., 2018. Mouse Motor Cortex Coordinates the Behavioral Response to Unpredicted Sensory Feedback. Neuron 0. https://doi.org/10.1016/j.neuron.2018.07.046Hong, S.Z., Mesik, L., Grossman, C.D., Cohen, J.Y., Lee, B., Severin, D., Lee, H.-K., Hell, J.W., Kirkwood, A., 2022. Norepinephrine potentiates and serotonin depresses visual cortical responses by transforming eligibility traces. Nat Commun 13, 3202. https://doi.org/10.1038/s41467-022-30827-1Kasamatsu, T., Pettigrew, J.D., 1976. Depletion of brain catecholamines: failure of ocular dominance shift after monocular occlusion in kittens. Science 194, 206–209. https://doi.org/10.1126/science.959850Lecas, J.-C., 2004. Locus coeruleus activation shortens synaptic drive while decreasing spike latency and jitter in sensorimotor cortex. Implications for neuronal integration. European Journal of Neuroscience 19, 2519–2530. https://doi.org/10.1111/j.0953-816X.2004.03341.xLeinweber, M., Ward, D.R., Sobczak, J.M., Attinger, A., Keller, G.B., 2017. A Sensorimotor Circuit in Mouse Cortex for Visual Flow Predictions. Neuron 95, 1420-1432.e5. https://doi.org/10.1016/j.neuron.2017.08.036Mahn, M., Prigge, M., Ron, S., Levy, R., Yizhar, O., 2016. Biophysical constraints of optogenetic inhibition at presynaptic terminals. Nat Neurosci 19, 554–556. https://doi.org/10.1038/nn.4266Margrie, T.W., Brecht, M., Sakmann, B., 2002. In vivo, low-resistance, whole-cell recordings from neurons in the anaesthetized and awake mammalian brain. Pflugers Arch. 444, 491–498. https://doi.org/10.1007/s00424-002-0831-zMcBurney-Lin, J., Lu, J., Zuo, Y., Yang, H., 2019. Locus coeruleus-norepinephrine modulation of sensory processing and perception: A focused review. Neurosci Biobehav Rev 105, 190–199. https://doi.org/10.1016/j.neubiorev.2019.06.009Müller, C.P., Pum, M.E., Amato, D., Schüttler, J., Huston, J.P., De Souza Silva, M.A., 2011. The in vivo neurochemistry of the brain during general anesthesia. Journal of Neurochemistry 119, 419–446. https://doi.org/10.1111/j.1471-4159.2011.07445.xNomura, S., Bouhadana, M., Morel, C., Faure, P., Cauli, B., Lambolez, B., Hepp, R., 2014. Noradrenalin and dopamine receptors both control cAMP-PKA signaling throughout the cerebral cortex. Front Cell Neurosci 8. https://doi.org/10.3389/fncel.2014.00247Polack, P.-O., Friedman, J., Golshani, P., 2013. Cellular mechanisms of brain-state-dependent gain modulation in visual cortex. Nat Neurosci 16, 1331–1339. https://doi.org/10.1038/nn.3464Raimondo, J.V., Kay, L., Ellender, T.J., Akerman, C.J., 2012. Optogenetic silencing strategies differ in their effects on inhibitory synaptic transmission. Nat Neurosci 15, 1102–1104. https://doi.org/10.1038/nn.3143Sakakibara, Y., Hirota, Y., Ibaraki, K., Takei, K., Chikamatsu, S., Tsubokawa, Y., Saito, T., Saido, T.C., Sekiya, M., Iijima, K.M., n.d. Widespread Reduced Density of Noradrenergic Locus Coeruleus Axons in the App Knock-In Mouse Model of Amyloid-β Amyloidosis. J Alzheimers Dis 82, 1513–1530. https://doi.org/10.3233/JAD-210385Sato, H., Fox, K., Daw, N.W., 1989. Effect of electrical stimulation of locus coeruleus on the activity of neurons in the cat visual cortex. Journal of Neurophysiology. https://doi.org/10.1152/jn.1989.62.4.946Seol, G.H., Ziburkus, J., Huang, S., Song, L., Kim, I.T., Takamiya, K., Huganir, R.L., Lee, H.-K., Kirkwood, A., 2007. Neuromodulators control the polarity of spike-timing-dependent synaptic plasticity. Neuron 55, 919–929. https://doi.org/10.1016/j.neuron.2007.08.013Shepard, K.N., Liles, L.C., Weinshenker, D., Liu, R.C., 2015. Norepinephrine is necessary for experience-dependent plasticity in the developing mouse auditory cortex. J Neurosci 35, 2432–2437.https://doi.org/10.1523/JNEUROSCI.0532-14.2015Vazey, E.M., Moorman, D.E., Aston-Jones, G., 2018. Phasic locus coeruleus activity regulates cortical encoding of salience information. Proceedings of the National Academy of Sciences 115, E9439– E9448. https://doi.org/10.1073/pnas.1803716115Yin, X., Jones, N., Yang, J., Asraoui, N., Mathieu, M.-E., Cai, L., Chen, S.X., 2021. Delayed motor learning in a 16p11.2 deletion mouse model of autism is rescued by locus coeruleus activation. Nat Neurosci 24, 646–657. https://doi.org/10.1038/s41593-021-00815-7